# Sheaf Neural Networks on SPD Manifolds:
# Second-Order Geometric Representation Learning

Yuhan Peng [* 1]   Junwen Dong [* 2]   Yuzhi Zeng [† 1]   Hao Li [† 3]   Ce Ju [4]   Huitao Feng [2]   Diaaeldin Taha [5 6]
Anna Wienhard [5 6]   Kelin Xia [1]

## Abstract

Graph neural networks face two fundamental challenges rooted in the linear structure of Euclidean vector spaces: (1) Current architectures represent geometry through vectors (directions, gradients), yet many tasks require matrix-valued representations that capture relationships between directions—such as how atomic orientations covary in a molecule. These second-order representations are naturally captured by points on the symmetric positive definite matrices (SPD) manifold; (2) Standard message passing applies shared transformations across edges. Sheaf neural networks address this via edge-specific transformations, but existing formulations remain confined to vector spaces and therefore cannot propagate matrix-valued features. We address both challenges by developing the first sheaf neural network operates natively on the SPD manifold. Our key insight is that the SPD manifold admits a Lie group structure, enabling well-posed analogs of sheaf operators without projecting to Euclidean space. Theoretically, we prove that SPD-valued sheaves are strictly more expressive than Euclidean sheaves: they admit consistent configurations (global sections) that vector-valued sheaves cannot represent, directly translating to richer learned representations. Empirically, our sheaf convolution transforms effectively rank-1 directional inputs into full-rank matrices encoding local geometric structure. Our dual-stream architecture achieves SOTA on 6/7 MoleculeNet benchmarks, with the sheaf framework providing consistent depth robustness.

## 1. Introduction

Consider predicting whether a drug molecule can penetrate the blood-brain barrier. Two molecules may share identical atoms and bonds leading to the same graph topology, yet differ dramatically in this property (Meng et al., 2021). What distinguishes them is often not *which* atoms connect, but *how* their bonds are oriented: the angles between adjacent bonds, the planarity of ring systems, the local anisotropy of molecular pockets—properties that describe relationships between directions, not directions themselves. A vector encodes a direction; capturing how directions *covary* requires a matrix. This is the difference between first-order and second-order geometric representations.

**Challenge 1: Limitation to first-order representations.** First-order representations correspond to vectors: directions, gradients, velocities. Second-order representations correspond to matrices, including covariances, curvatures, and Hessians, capturing *relationships between directions*. These second-order representations are captured by symmetric matrices, which encode additional geometric structure that vectors alone do not represent. When positive definite, these matrices are points on $\mathrm{SPD}_n$, the manifold of symmetric positive definite matrices, which is a Riemannian manifold of non-positive curvature.

Current GNNs are confined to first-order representations. Despite recent progress in geometric architectures—whether through angle encodings (Gasteiger et al., 2020; 2021; Liu et al., 2022b), directional message passing (Schütt et al., 2021; Satorras et al., 2021), or spherical harmonics (Thomas et al., 2018)—these methods all represent geometry through scalars, vectors, never through matrices directly. Even methods designed for SPD data resort to a "Euclidean detour" (Huang & Gool, 2017; Brooks et al., 2019; Wang et al., 2022a): projecting to the tangent space to perform Euclidean operations before mapping back. While computationally convenient, such operations inevitably discard

[*]Equal contribution [†]Equal contribution [1]School of Physical and Mathematical Sciences, Nanyang Technological University, Singapore [2]Chern Institute of Mathematics, Nankai University, Tianjin, China [3]Department of Mathematics, Faculty of Science, National University of Singapore, Singapore [4]Inria, CEA, Université Paris-Saclay, Palaiseau, France [5]Max Planck Institute for Mathematics in the Sciences, Leipzig, Germany [6]Center for Scalable Data Analytics and Artificial Intelligence, Leipzig, Germany. Correspondence to: Kelin Xia <xiakelin@ntu.edu.sg>, Anna Wienhard <anna.wienhard@mis.mpg.de>, Diaaeldin Taha <diaaeldin.taha@mis.mpg.de>.

*Proceedings of the 43rd International Conference on Machine Learning*, Seoul, South Korea. PMLR 306, 2026. Copyright 2026 by the author(s).

intrinsic geometric information and therefore fail to fully exploit the representational power of the SPD manifold. We seek instead to work natively on $\mathrm{SPD}_n$ (López et al., 2021).

**Challenge 2: Euclidean constraints on sheaf convolution.** Standard message passing aggregates neighbors through a shared transformation—implicitly assuming that information should combine the same way along every edge. This homogeneity assumption fails when relationships are heterogeneous: a molecule's rigid ring bonds behave differently from its flexible chain bonds. Sheaf neural networks (Hansen & Gebhart, 2020; Barbero et al., 2022) address this by equipping each edge with a *restriction map* that encodes edge-specific transformations. However, existing formulations operate only on vector spaces; extending sheaves to non-Euclidean geometries remains unexplored. Consequently, while sheaves provide a principled way to model heterogeneity, they cannot natively propagate matrix-valued, second-order representations.

**Connecting the challenges.** These limitations share a common mathematical origin: the linear structure of Euclidean vector spaces. Vector spaces lack capacity to encode second-order representations, and sheaf constructions with linear operators cannot be directly lifted to the non-Euclidean manifolds where such representations naturally reside.

**Our solution: SPD-valued sheaves.** We resolve both challenges by developing the first sheaf framework natively on the SPD manifold (Figure 1). The central difficulty is that the SPD manifold is not a vector space, so basic operations such as coboundary operators and Laplacian operators do not admit their standard linear definitions. To address this, we use the Lie group structure induced by the global diffeomorphism $\log : \mathrm{SPD}_n \to \mathrm{Sym}_n$ (the vector space of $n \times n$ real symmetric matrices) and develop Riemannian counterparts of sheaf-theoretic operators through the resulting $\log/\exp$-induced algebraic structure. From the viewpoint of discrete fiber bundles, this construction should be distinguished from initially projecting all SPD inputs into a single Euclidean space via the logarithm. Although both formulations involve $\log/\exp$ calculations algebraically, their geometric interpretations differ. A Euclidean projection globally flattens all SPD inputs into a single Euclidean space and corresponds to a flat bundle, whereas our construction uses the $\log/\exp$-induced linear structure locally within SPD-valued fibers and can be viewed as operating over a non-flat discrete bundle. This enables geometry-aware message passing that respects both SPD's Riemannian structure and sheaf theory's edge-dependent transformations.

**Contributions.** We propose the first principled framework for geometric deep learning with second-order representations via SPD sheaves. Our contributions are three-fold:

1. **Mathematical Framework**: We extend sheaf framework to SPD manifold, defining isometric restriction maps and a coboundary operator compatible with the

Lie group structure. This enables the sheaf Laplacian to act on $\mathrm{SPD}_n$ while avoiding the global flattening of SPD inputs into a single Euclidean feature space.

2. **Theoretical Analysis**: We prove that SPD sheaves strictly generalize Euclidean sheaves, admitting full-rank matrix representations that Euclidean stalks cannot express (Theorem 4.15).

3. **Empirical Validation**: Our dual-stream architecture achieves state-of-the-art on 6 of 7 MoleculeNet datasets. We observe that sheaf convolution induces *second-order emergence*: effectively rank-1 initializations evolve into full-rank matrices, encoding local geometric structure. Our architecture also exhibits depth robustness, retaining 97% performance at 32 layers.

## 2. Related Work

**Geometric molecular representations.** Message passing neural networks (Gilmer et al., 2017) provide a unified framework for molecular learning. Incorporating 3D structure drove substantial progress through interatomic distances (Schütt et al., 2017), bond angles (Gasteiger et al., 2020; Liu et al., 2022b), and dihedral angles (Gasteiger et al., 2021).

Equivariant architectures (Thomas et al., 2018; Fuchs et al., 2020; Satorras et al., 2021; Schütt et al., 2021) build symmetry constraints directly into the model. All these methods encode geometry through scalars (distances, angles) or vectors (directions, displacements)—fundamentally first-order representations. Even tensor-product constructions (Thomas et al., 2018) producing matrix-like objects operate through iterative vector-level transformations, without directly exploiting the geometric structure of matrix manifolds such as SPD, motivating our SPD-based approach.

**Second-order representations and SPD matrix learning.** Second-order statistics capture richer information than first-order features: covariance pooling has demonstrated gains in visual recognition (Li et al., 2017; Wang et al., 2020) , and SPD matrices arise naturally in diffusion tensor imaging, radar signal processing, and molecular geometry. SPDNet (Huang & Gool, 2017) pioneered deep learning on SPD manifolds, followed by Riemannian batch normalization (Brooks et al., 2019), and gyrovector space formulations (López et al., 2021; Nguyen & Yang, 2023). Recent work has extended SPD matrix representations to graphs. Zhao et al. (2023) proposed a Log-Euclidean framework operating in the tangent space, while Wang & Chang (2026) operate directly on $\mathrm{SPD}_n$ via Cholesky decomposition, avoiding the tangent-space projection. Yet two limitations remain: (i) their Euclidean-to-SPD embedding is learned via multilayer perceptrons (MLPs), lacking explicit geometric meaning, whereas we propose a geometry-aware lifting $\hat{u}_v \hat{u}_v^\top + \varepsilon I$ that preserves directional structure without learning; (ii) they employ standard message passing

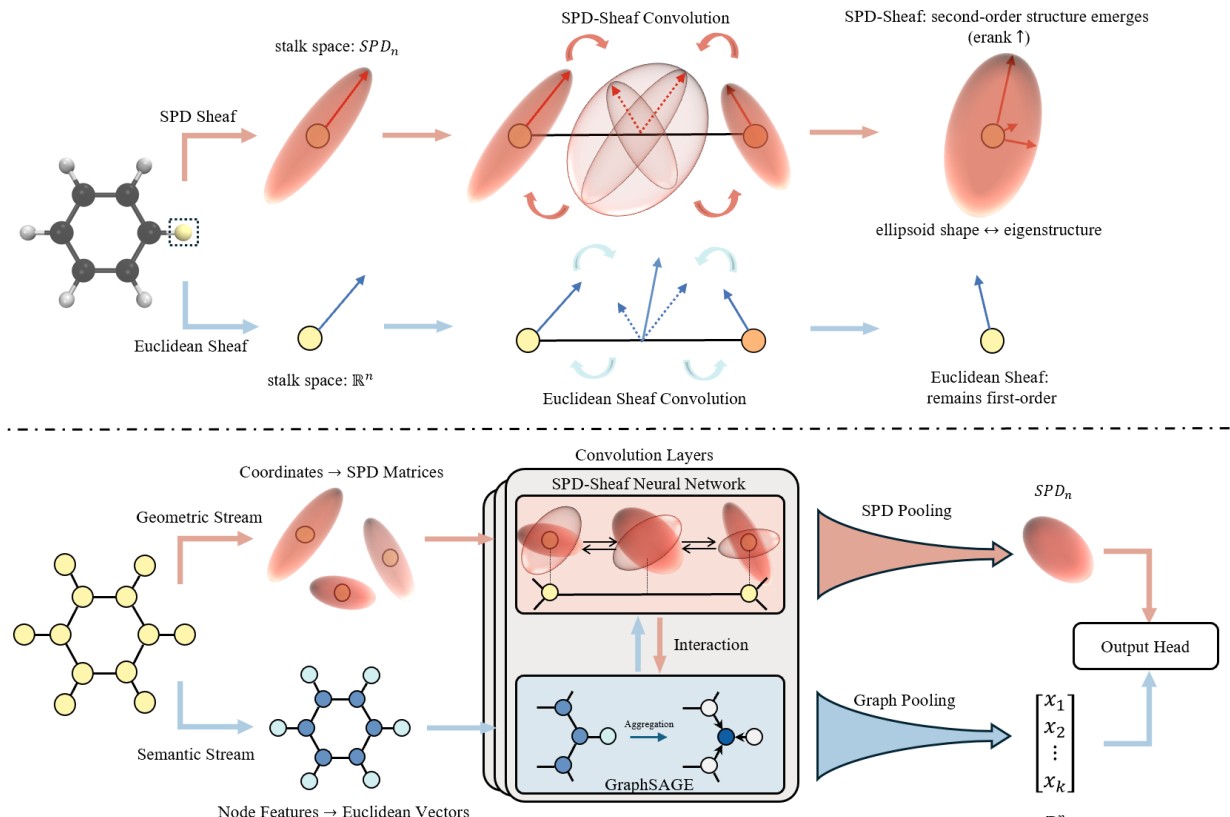

*Figure 1.* **SPD Sheaf Neural Networks.** *Top:* SPD sheaves assign matrix-valued stalks ($\text{SPD}_n$) rather than vector stalks ($\mathbb{R}^n$). Through sheaf convolution, SPD representations gain second-order structure (effective rank increases), while Euclidean representations remain first-order. *Bottom:* Dual-stream architecture: the geometric stream processes coordinates via SPD sheaf convolution; the semantic stream processes node features via GraphSAGE, with cross-modal interaction enabling information exchange between streams.

without edge-specific transformations, limiting expressivity on heterophilic graphs. Our framework addresses both gaps by combining geometrically meaningful SPD embeddings with the sheaf neural network described below.

**Cellular sheaves and sheaf neural networks.** Cellular sheaves generalize graphs by attaching vector spaces (stalks) to cells, connected by linear restriction maps (Curry, 2014; Hansen & Ghrist, 2019). Hansen & Gebhart (2020) introduced sheaf neural networks, learning restriction maps to capture edge-heterogeneous relationships. Neural Sheaf Diffusion (Bodnar et al., 2022) demonstrated that sheaf structure addresses heterophily, while subsequent work explored connection Laplacians (Barbero et al., 2022) and nonlinear extensions (Zaghen, 2024). However, existing sheaf networks assume Euclidean stalks: restriction maps must be linear, and the coboundary operator relies on vector subtraction unavailable on manifolds. Our contribution extends sheaf theory to SPD stalks, enabling convolution of second-order geometric information with edge-specific transformations. As a byproduct, the sheaf Laplacian's edge-dependent structure also mitigates over-smoothing (Oono & Suzuki, 2020; Bodnar et al., 2022), a benefit we verify

empirically in Section 6.5.

## 3. Background

### 3.1. The SPD Manifold

**Definition 3.1** (SPD matrices). *A matrix $X \in \mathbb{R}^{n \times n}$ is symmetric positive definite* (SPD) *if $X = X^\top$ and $x^\top X x > 0$ for all $x \in \mathbb{R}^n \setminus \{0\}$. The space of $n \times n$ SPD matrices forms a smooth manifold denoted $\text{SPD}_n$, which can be identified with the open cone in the $\frac{n(n+1)}{2}$-dimensional space of symmetric matrices.*

**Riemannian structure.** The *affine-invariant Riemannian metric* (AIRM), a widely adopted metric on $\text{SPD}_n$, is given by $g_P^{\text{AI}}(V, U) := \text{tr}(P^{-1}UP^{-1}V)$, where $V, U \in T_P\text{SPD}_n$. Another commonly used metric is the Log-Euclidean metric, defined as $g_P^{\text{LEM}}(V, U) := \langle D_P \log V, \ D_P \log U \rangle_F$, where $D_P \log$ denotes the differential of the matrix logarithm at $P$.

**Lemma 3.2.** *Under both the affine-invariant and Log-Euclidean metrics, $\text{SPD}_n$ satisfies the following properties:*

1. *$\text{SPD}_n$ is geodesically complete.*
2. *The Riemannian exponential and logarithm at the*

*identity matrix $I_n$ coincide with the matrix exponential and logarithm i.e., $\exp_{I_n}(X) = \exp(X)$ and $\log_{I_n}(P) = \log(P)$.*

Although the affine-invariant and Log-Euclidean metrics induce substantially different global geometries on $\mathrm{SPD}_n$, the constructions developed in this paper rely only on the properties listed in Lemma 3.2. Consequently, all results hold under either choice of metric.

**Lie group structure.** Beyond its Riemannian geometry, $\mathrm{SPD}_n$ admits a natural Lie group structure that plays a central role in our sheaf construction. Fixing the identity matrix $I_n$ as base point, the logarithm $\log : \mathrm{SPD}_n \longrightarrow \mathrm{Sym}_n$ provides a global diffeomorphism. This enables transporting the additive group structure of $\mathrm{Sym}_n$ to $\mathrm{SPD}_n$ and defining a binary operation

$$P \odot Q := \exp(\log P + \log Q). \qquad (1)$$

Under this operation, $(\mathrm{SPD}_n, \odot)$ forms an Abelian Lie group with identity element $I_n$ and inverse given by $P^{-1} = \exp(-\log P)$.

As established in Lemma 3.2, in view of the geodesic completeness of $\mathrm{SPD}_n$ and the compatibility between the matrix and Riemannian logarithm and exponential at the identity under both the affine-invariant and Log-Euclidean metrics, the Lie group operation $\odot$ admits the following interpretation: any point $P \in \mathrm{SPD}_n$ can be mapped to the common tangent space $T_{I_n}\mathrm{SPD}_n$ via the Riemannian logarithm at $I_n$, group addition is performed in this linear space, and the result is mapped back to the manifold via the Riemannian exponential at $I_n$.

### 3.2. Cellular Sheaves on Graphs

We recall the classical cellular sheaf theory on graphs (Hansen & Gebhart, 2020). This provides the algebraic framework that we will generalize to SPD-valued stalks.

**Definition 3.3** (Cellular Sheaf). A *cellular sheaf* $(G, \mathcal{F})$ on an undirected graph $G = (V, E)$ consists of:

- A vector space $\mathcal{F}(v)$ (the *stalk*) for each vertex $v \in V$;
- A vector space $\mathcal{F}(e)$ for each edge $e \in E$;
- A linear *restriction map* $\mathcal{F}_{v \to e} : \mathcal{F}(v) \to \mathcal{F}(e)$ for every incident pair $v \in e$.

The restriction maps encode edge-specific transformations, enabling sheaves to capture heterogeneous relationships.

**Cochain spaces and coboundary operator.** The space of *0-cochains* is $C^0(G, \mathcal{F}) := \bigoplus_{v \in V} \mathcal{F}(v)$, and the space of *1-cochains* is $C^1(G, \mathcal{F}) := \bigoplus_{e \in E} \mathcal{F}(e)$. The *coboundary operator* $\delta : C^0 \to C^1$ measures local inconsistency:

$$\delta(\mathbf{x})_e = \mathcal{F}_{v \to e}\mathbf{x}_v - \mathcal{F}_{u \to e}\mathbf{x}_u, \qquad (2)$$

for each edge $e = (u, v)$ with a chosen orientation. Note that this definition relies on vector subtraction, which is a structure unavailable on general manifolds.

**Sheaf Laplacian.** The *sheaf Laplacian* $\mathbf{L}_\mathcal{F} : C^0 \to C^0$ is defined by $\delta^\top \delta$. The kernel $\ker \mathbf{L}_\mathcal{F} = \ker \delta$ consists of *global sections*, the 0-cochains that are consistent across all edges. By the sheaf-theoretic Hodge decomposition, these correspond to the harmonic cochains and determine the representational capacity of sheaf convolutions.

**Sheaf neural networks.** Sheaf neural networks (Bodnar et al., 2022) use the sheaf Laplacian for message passing. Let $\mathbf{H}^{(\ell)} \in \mathbb{R}^{|V|d \times f}$ denote the node feature matrix at layer $\ell$, stacking the stalk-valued representations of all vertices, and $\mathbf{W}^{(\ell)} \in \mathbb{R}^{f \times f'}$ a learnable channel-mixing weight matrix. The update rule is

$$\mathbf{H}^{(\ell+1)} = \sigma\left( (I - \mathbf{L}_\mathcal{F})\mathbf{H}^{(\ell)}\mathbf{W}^{(\ell)} \right), \qquad (3)$$

where the restriction maps are learned. This enables edge-specific transformations while maintaining a convolution framework. However, the stalks $\mathcal{F}(v)$ remain vector spaces, limiting representations to first-order information.

## 4. SPD-Valued Sheaf Theory

We now develop the theory of sheaves with stalks valued in $\mathrm{SPD}_n$. This construction addresses both challenges identified in the introduction simultaneously: SPD-valued stalks enable native processing of second-order geometric information (Challenge 1), while the Lie group structure of $\mathrm{SPD}_n$ allows us to extend the sheaf framework beyond Euclidean stalks, defining coboundary operators and Laplacians that respect manifold geometry (Challenge 2). All proofs in this section are deferred to Appendix A.

### 4.1. SPD-Valued Sheaves and Restriction Maps

The first step in extending sheaf theory to SPD manifolds is to define appropriate restriction maps. In classical sheaves, restriction maps are linear transformations between vector spaces. For SPD-valued sheaves, we require well-defined maps from $\mathrm{SPD}_n$ to $\mathrm{SPD}_n$ and respect the intrinsic geometric structure of the manifold.

**Definition 4.1** (SPD Sheaf). An *SPD sheaf* $(G, \mathcal{F})$ on a graph $G = (V, E)$ assigns $\mathcal{F}(v) = \mathrm{SPD}_n$ to each vertex $v$ and $\mathcal{F}(e) = \mathrm{SPD}_n$ to each edge $e$. For each incident pair $v \in e$, the restriction map is:

$$\mathcal{F}_{v \to e}(P) = M_{ve}PM_{ve}^\top, \quad M_{ve} \in O(n), \qquad (4)$$

where $O(n)$ denotes the orthogonal group.

The choice of orthogonal restriction maps is canonical:

**Proposition 4.2** (Restriction Maps are Isometries). *Let $g$ denote either the affine-invariant Riemannian metric $g^{AIRM}$ or the Log-Euclidean metric $g^{LEM}$ on $\mathrm{SPD}_n$. For any $M \in O(n)$, the map $\phi_M : P \mapsto MPM^\top$ is an isometry of $(\mathrm{SPD}_n, g)$.*

The orthogonal group $O(n)$ fixes the base point $I_n$ and is the maximal subgroup of $GL(n)$ whose congruence action preserves both AIRM and Log-Euclidean metric (Thanwerdas & Pennec, 2023). This parallels the Euclidean case: just as orthogonal matrices preserve the Euclidean metric on $\mathbb{R}^n$, orthogonal congruence transformations preserve the metric on $\mathrm{SPD}_n$. This ensures that edge-specific transformations, which in classical sheaf theory require Euclidean structure, can now operate natively within SPD geometry, thereby addressing Challenge 2.

## 4.2. The Coboundary Operator

The classical coboundary operator $\delta(\mathbf{x})_e = \mathcal{F}_{v \to e}\mathbf{x}_v - \mathcal{F}_{u \to e}\mathbf{x}_u$ relies fundamentally on vector subtraction. Since $\mathrm{SPD}_n$ is not a vector space, we must reformulate this construction. The key is to exploit the Lie group structure: subtraction in $\mathbb{R}^n$ becomes subtraction in the Lie algebra $\mathrm{Sym}_n$, followed by the exponential map back to the group.

**Definition 4.3** (SPD Coboundary Operator). For a 0-cochain $\sigma = (\sigma_v)_{v \in V}$ with $\sigma_v \in \mathrm{SPD}_n$, the coboundary $\delta\sigma \in C^1$ is defined on each oriented edge $e = (u, v)$ as:

$$(\delta\sigma)_e = \exp\left(\log(\mathcal{F}_{v \to e}(\sigma_v)) - \log(\mathcal{F}_{u \to e}(\sigma_u))\right). \quad (5)$$

This definition directly parallels the Euclidean case and preserves the essential algebraic property:

**Proposition 4.4** (Linearity of Coboundary). *The coboundary operator $\delta$ is linear with respect to the Lie group operation $\odot$: $\delta(\sigma \odot \tau)_e = (\delta\sigma)_e \odot (\delta\tau)_e$.*

## 4.3. The Adjoint Operator and Sheaf Laplacian

In classical sheaf theory, the adjoint $\delta^\top$ is defined via the Euclidean inner product on stalks. In contrast, $\mathrm{SPD}_n$ carries no canonical inner product. However, this is not an obstacle: the definition of an adjoint requires only a non-degenerate bilinear pairing. We therefore introduce an SPD pairing and define $\delta^\top$ through the corresponding Green identity; details are given in Appendix A.

**Definition 4.5** (SPD Pairing). The pairing on $\mathrm{SPD}_n$ is defined as:

$$\langle X, Y \rangle := g_{I_n}(\log X, \log Y). \quad (6)$$

where $g_{I_n}$ is the inner product on the tangent space $T_{I_n}\mathrm{SPD}_n$ induced by the metric $g$ at the point $I_n$.

This pairing is compatible with the Lie group structure: $\langle X, Y_1 \odot Y_2 \rangle = \langle X, Y_1 \rangle + \langle X, Y_2 \rangle$.

**Proposition 4.6** (Adjoint Operator). *Fix an arbitrary orientation for each edge $e = (u, v)$. To encode the resulting orientation-dependent signs, we introduce an incidence index $I(v, e) \in \{0, 1\}$, defined as follows: if $e$ is oriented as $v \to u$, then $I(v, e) = 0$ and $I(u, e) = 1$.*

*The adjoint $\delta^\top : C^1 \to C^0$ satisfying $\langle \delta\sigma, \tau \rangle = \langle \sigma, \delta^\top\tau \rangle$ is given by:*

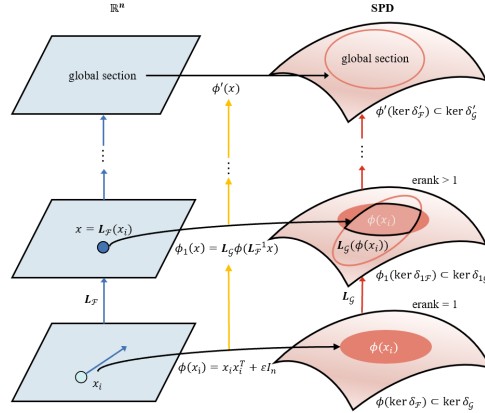

*Figure 2.* **Relationship to Euclidean Sheaves.** Left: Laplacian-based updates on a Euclidean sheaf $\mathcal{F}$. Right: updates on the corresponding SPD-valued sheaf $\mathcal{G}$. The embedding $\Phi$ lifts Euclidean features to SPD matrices, satisfying $\Phi(\ker \delta_{\mathcal{F}}) \subseteq \ker \delta_{\mathcal{G}}$. This inclusion is preserved across layers, with final representations corresponding to global sections. The strict inclusion highlights the greater representational capacity of SPD-valued sheaves.

$$(\delta^\top \tau)_v = \exp\left(\sum_{v \to e}(-1)^{I(v,e)} M_{ve}^\top (\log \tau_e) M_{ve}\right), \quad (7)$$

**Definition 4.7** (SPD Sheaf Laplacian). The SPD sheaf Laplacian $\mathbf{L}_\mathcal{F} := \delta^\top \circ \delta$ acts on SPD-valued cochains as:

$$(\mathbf{L}_\mathcal{F} X)_v = \exp\left(\sum_{(v,u)=e}(-1)^{I(v,e)} M_{ve}^\top \left(\log(\mathcal{F}_{v \to e} X_v)\right.\right.$$
$$\left.\left. - \log(\mathcal{F}_{u \to e} X_u)\right) M_{ve}\right). \quad (8)$$

**Proposition 4.8** (Hodge-type Decomposition for SPD-valued Sheaves). *With the coboundary operator $\delta$, its adjoint $\delta^\top$, and the Laplacian $\mathbf{L} = \delta^\top \delta$ defined above, harmonic 0-cochains coincide with global sections:*

$$\ker \mathbf{L} = \ker \delta. \quad (9)$$

*Remark* 4.9 (Edge Heterogeneity on SPD Manifolds). The relative transformation from $\mathcal{F}_u$ to $\mathcal{F}_v$ on edge $e = u \to v$ defined by $\mathcal{F}_u \xrightarrow{\mathcal{F}_{u \to e}} \mathcal{F}_e \xleftarrow{\mathcal{F}_{v \to e}} \mathcal{F}_u$ captures edge-specific relationships, enabling the sheaf Laplacian to model heterogeneous inter-node interactions. When all transformations are identity, the SPD sheaf Laplacian reduces to the standard graph Laplacian on matrix-valued signals. This edge-specificity also prevents over-smoothing, as verified in Section 6.5.

## 4.4. Relationship to Euclidean Sheaves

Having constructed the SPD sheaf framework, we now establish its relationship to classical Euclidean sheaves. This clarifies in what sense SPD sheaves constitute a *generalization* rather than merely an alternative formulation.

**Definition 4.10** (Euclidean-to-SPD Embedding). For $\varepsilon > 0$, define $\Phi_\varepsilon : \mathbb{R}^n \to \mathrm{SPD}_n$ by:

$$\Phi_\varepsilon(x) = xx^\top + \varepsilon I_n. \tag{10}$$

This embedding lifts first-order vectors to second-order representations (effective rank-1 SPD matrices).

**Proposition 4.11** (Restriction Maps as Pullbacks). *Each $\Sigma \in \mathrm{SPD}_n$ defines an inner product $\langle x, y \rangle_\Sigma = x^\top \Sigma y$ on $\mathbb{R}^n$. Under this identification, SPD restriction maps are induced from the transposes of the Euclidean restriction maps via pullback:*

$$\mathcal{G}_{v \to e} := (\mathcal{F}_{v \to e}^\top)^*, \tag{11}$$

*where $(\mathcal{F}_{v \to e}^\top)^* \Sigma(x, y) := \Sigma(\mathcal{F}_{v \to e}^\top x, \mathcal{F}_{v \to e}^\top y)$.*

*Remark* 4.12 (Naturality of Orthogonal Restriction Maps). The choice of orthogonal congruence as SPD restriction maps is structurally determined. Consider the embedding $\Phi$ in (10): if a Euclidean restriction map acts as $x \mapsto Mx$ with $M \in O(n)$, then $\Phi(Mx) = (Mx)(Mx)^\top + \varepsilon I = M\Phi(x)M^\top$. Thus, the congruence action $P \mapsto MPM^\top$ on $\mathrm{SPD}_n$ is the canonical lift of restriction maps on $\mathbb{R}^n$ when extending Euclidean sheaves to SPD-valued sheaves.

**Proposition 4.13** (Kernel Correspondence). *Let $\mathcal{F}$ be a Euclidean sheaf and $\mathcal{G}$ an SPD sheaf with matched restriction maps. Then:*

$$\delta_\mathcal{F} \mathbf{x} = 0 \Rightarrow \delta_\mathcal{G}(\Phi(\mathbf{x})) = I_n. \tag{12}$$

*That is, the embedding preserves global sections. On the other side, the embedding global section feedback as $\delta_\mathcal{G}(\Phi(\mathbf{x})) = I_n \Rightarrow \delta_\mathcal{F}(|\mathbf{x}|) = 0$.*

*Remark* 4.14. For graph with $\mathbb{Z}_2$-gauge ambiguity, especially trees, the feedback equation $\delta_\mathcal{F}(|\mathbf{x}|) = 0$ reduces to $\delta_\mathcal{F} \mathbf{x} = 0$, meaning the image of global section feedback lies in global sections. This shows SPD-sheaves preserve geometric capacity by quotienting out line-bundle frustrations that Euclidean sheaves cannot process. Moreover, $\ker \delta_\mathcal{G}$ is "larger" than $\ker \delta_\mathcal{F}$, as formalized below.

**Theorem 4.15** (Strict Generalization). *Let $\mathcal{F}$ be a Euclidean cellular sheaf and $\mathcal{G}$ its associated SPD-valued sheaf induced by the embedding $\Phi$, with corresponding Laplacians $\mathbf{L}_\mathcal{F}$ and $\mathbf{L}_\mathcal{G}$. Then $\Phi$ preserves global sections:*

$$\Phi(\ker \mathbf{L}_\mathcal{F}) \subsetneq \ker \mathbf{L}_\mathcal{G}. \tag{13}$$

*That is, the SPD sheaf admits strictly more global sections than the Euclidean sheaf, illustrated in Figure 2.*

The strict inclusion can be understood quantitatively through a comparison of their discrete indices. We show that the enlargement of the harmonic space for SPD-valued sheaves is governed by an explicit index jump.

*Remark* 4.16 (Representational Capacity). By sheaf-theoretic Hodge decomposition, $\ker \mathbf{L}_\mathcal{F}$ corresponds to global sections. For standard graph Laplacians on connected graphs, this space is typically one-dimensional, leading to over-smoothing at steady states. Sheaf neural networks mitigate this collapse via non-trivial restriction maps. Moreover, Theorem 4.15 shows that SPD sheaves strictly expand the kernel beyond Euclidean sheaves. This implies that while Euclidean and SPD frameworks resist over-smoothing, SPD sheaves preserve a fundamentally richer space of global sections. Consequently, the signal maintained at depth encodes complex second-order geometric configurations that vector-valued sheaves cannot fundamentally represent.

## 5. Model Architecture

We instantiate SPD sheaf theory as a neural architecture. This applies to any domain where SPD matrices arise naturally (e.g., EEG covariance matrices; see Appendix E). We demonstrate on molecular property prediction, lifting 3D coordinates to SPD matrices via geometry-aware embedding. Our design follows from the theoretical development: geometric and semantic information possess different mathematical structures and should be processed accordingly.

### 5.1. Dual-Stream Design

Molecular graphs encode two distinct information types:

- **Geometric structure**: 3D atomic coordinates define directional relationships naturally captured by SPD matrices encoding local geometric structure.
- **Chemical semantics**: Atom types are discrete, categorical quantities represented in Euclidean space.

This motivates a dual-stream architecture where a *geometric stream* processes coordinates via SPD sheaf convolution, while a *semantic stream* handles chemical features via standard graph neural networks as shown in Figure 1 bottom.

**Coordinate-to-SPD lifting.** Given atomic coordinates $\{p_v\}_{v \in V}$, we compute centered displacement vectors $u_v = p_v - \bar{p}$ where $\bar{p}$ is the molecular centroid and $\hat{u}_v = \frac{u_v}{\|u_v\| + \epsilon}$, then lift to initial effectively rank-1 SPD matrices:

$$X_v^{(0)} = \hat{u}_v \hat{u}_v^\top + \varepsilon I_3 \in \mathrm{SPD}_3, \tag{14}$$

To enable learnable geometric transformations while preserving SPD structure, we apply a transformation: $\tilde{X}_v^{(\ell)} = Q_v^{(\ell)} X_v^{(\ell)} Q_v^{(\ell)\top}$, where $Q_v^{(\ell)} \in O(3)$ is learnable.

**SPD sheaf convolution.** The geometric stream applies the SPD sheaf Laplacian with Lie group updates. Given node representations $X_v^{(\ell)} \in \mathrm{SPD}_3$, the update rule is:

$$X_v^{(\ell+1)} = X_v^{(\ell)} \odot \left( \mathbf{L}_\mathcal{F}^{(\ell)} \tilde{X}^{(\ell)} \right)_v, \tag{15}$$

where the Laplacian term follows Definition 4.7 with learned restriction maps $M_{ve}^{(\ell)}, M_{ue}^{(\ell)} \in O(3)$. Specifically, adjacent

node features are concatenated and passed through an MLP, then antisymmetrized to obtain skew-symmetric matrices $S$, which are converted to orthogonal matrices via the scaled Cayley transform $M = (I - S/2)^{-1}(I + S/2)$, ensuring exact orthogonality: $M_{ue}^{(\ell)} = \text{Cayley}\left(\text{MLP}([h_u^{(\ell)}\|h_v^{(\ell)}])\right)$. This design allows the restriction maps to adapt to local chemical environments while maintaining the isometry property required by Proposition 4.2. A TgReEig nonlinearity (Zhao et al., 2023) is applied after each update.

**Semantic stream.** Chemical features are processed via GraphSAGE (Hamilton et al., 2017) with mean aggregation:

$$h_v^{(\ell+1)} = \sigma\left(W^{(\ell)} \cdot \text{CONCAT}\left(h_v^{(\ell)}, \text{MEAN}_{u \in \mathcal{N}(v)} h_u^{(\ell)}\right)\right).$$

### 5.2. Cross-Modal Interaction

While the streams process different modalities, molecular properties depend on their joint information. We introduce asymmetric cross-modal interaction where geometry modulates semantics: $h_v^{(\ell)} \leftarrow h_v^{(\ell)} + \alpha_v \cdot W_\xi \text{vec}(\log X_v^{(\ell)})$,

where $\alpha_v = \sigma(\text{MLP}([W_{\text{spd}}[\text{vec}(\log X_v^{(\ell)})\|W_{\text{feat}}h_v^{(\ell)}])$ is a learned attention weight. This preserves geometric integrity while allowing geometry to inform semantic processing.

### 5.3. Pooling and Prediction

**SPD pooling.** For graph-level prediction, we aggregate node SPD matrices via power-Euclidean pooling:

$$\bar{X}_G = \left(\frac{1}{|V|}\sum_{v \in V}(X_v^{(L)})^\theta\right)^{1/\theta}, \qquad (16)$$

$\theta \in (0, 1]$. This pooling is computed in closed form and guarantees the output remains in $\text{SPD}_n$ (Chen et al., 2025).

**Output head: fusion and prediction.** The pooled SPD matrix is vectorized as $g = \text{vec}(\log \bar{X}_G)$. The semantic stream produces $h_G$ via mean pooling. We fuse via factorized bilinear pooling: $z = (Ug) \circ (Vh_G)$, $\hat{y} = \text{MLP}(\text{BN}([g\|h_G\|z]))$. Further implementation details are provided in Appendix D.

## 6. Experiments and Results

Our experiments are designed to validate the two main empirical claims from the introduction: (1) SPD representations provide substantial gains on geometry-sensitive tasks by capturing second-order information (Challenge 1), and (2) our extension of sheaf neural network to SPD manifolds enables edge-specific transformations on second-order features, providing depth robustness (Challenge 2). We further analyze the geometric mechanism underlying second-order emergence in SPD convolution.

*Table 1.* Comparison with methods on MoleculeNet benchmarks. We report ROC-AUC (%) with standard deviations as subscripts. **Bold** indicates the best result; underline indicates second-best.

| Model | BACE | BBBP | ClinTox | SIDER | Tox21 | HIV | MUV |
|---|---|---|---|---|---|---|---|
| No. molecules | 1,513 | 2,039 | 1,478 | 1,427 | 7,831 | 41,127 | 93,808 |
| No. avg atoms | 65 | 46 | 50.58 | 65 | 36 | 46 | 43 |
| No. tasks | 1 | 1 | 2 | 27 | 12 | 1 | 17 |
| D-MPNN | $80.9_{0.6}$ | $71.0_{0.3}$ | $90.6_{0.7}$ | $57.0_{0.7}$ | $75.9_{0.7}$ | $77.1_{0.5}$ | $78.6_{1.4}$ |
| AttentiveFP | $78.4_{2.2}$ | $66.3_{1.8}$ | $84.7_{0.3}$ | $60.6_{3.2}$ | $78.1_{0.5}$ | $75.7_{1.4}$ | $78.6_{1.5}$ |
| N-GramRF | $77.9_{1.5}$ | $69.7_{0.6}$ | $77.5_{4.0}$ | $66.8_{0.7}$ | $74.3_{0.9}$ | $77.2_{0.4}$ | $76.9_{0.2}$ |
| N-GramXGB | $79.1_{1.3}$ | $69.1_{0.8}$ | $87.5_{2.7}$ | $65.5_{0.7}$ | $75.8_{0.9}$ | $78.7_{0.4}$ | $74.8_{0.2}$ |
| PretrainGNN | $84.5_{0.7}$ | $68.7_{1.3}$ | $72.6_{1.5}$ | $62.7_{0.8}$ | $78.1_{0.6}$ | $79.9_{0.7}$ | $81.3_{2.1}$ |
| GROVE$_{\text{base}}$ | $82.1_{0.7}$ | $70.0_{0.1}$ | $81.2_{3.0}$ | $64.8_{0.6}$ | $74.3_{0.1}$ | $62.5_{0.9}$ | $67.3_{1.8}$ |
| GROVE$_{\text{large}}$ | $81.0_{1.4}$ | $69.5_{0.1}$ | $76.2_{3.7}$ | $65.4_{0.1}$ | $73.5_{0.1}$ | $68.2_{1.1}$ | $67.3_{1.8}$ |
| GraphMVP | $81.2_{0.9}$ | $72.4_{1.6}$ | $79.1_{2.8}$ | $63.9_{1.2}$ | $75.9_{0.5}$ | $77.0_{1.2}$ | $77.7_{0.6}$ |
| MolCLR | $82.4_{0.9}$ | $72.2_{2.1}$ | $91.2_{3.5}$ | $58.9_{1.4}$ | $75.0_{0.2}$ | $78.1_{0.5}$ | $79.6_{1.9}$ |
| GEM | $85.6_{1.1}$ | $72.4_{0.4}$ | $90.1_{1.3}$ | $67.2_{0.4}$ | $78.1_{0.1}$ | $80.6_{0.9}$ | $81.7_{0.5}$ |
| Mol-GDL | $86.3_{1.9}$ | $72.8_{1.9}$ | $96.6_{0.2}$ | $83.1_{0.2}$ | $79.4_{0.5}$ | $80.8_{0.7}$ | $67.5_{1.4}$ |
| Uni-Mol | $85.7_{0.2}$ | $72.9_{0.6}$ | $91.9_{1.8}$ | $65.9_{1.3}$ | $79.6_{0.5}$ | $80.8_{0.7}$ | $82.1_{1.3}$ |
| SMPT | $87.3_{1.5}$ | $73.4_{0.3}$ | $92.7_{0.2}$ | $67.6_{5.0}$ | $79.7_{0.1}$ | **$81.2_{0.1}$** | **$82.2_{0.8}$** |
| SchNet | $73.7_{0.9}$ | $70.8_{1.4}$ | $97.8_{0.3}$ | $83.1_{0.0}$ | $72.4_{0.2}$ | $68.8_{0.1}$ | $68.2_{0.6}$ |
| EGNN | $79.4_{2.3}$ | $72.1_{2.0}$ | $98.6_{0.5}$ | $83.2_{0.1}$ | $74.5_{0.3}$ | $70.4_{0.2}$ | $68.6_{0.5}$ |
| **SPD-Sheaf (Ours)** | **$89.0_{1.4}$** | **$77.4_{2.7}$** | **$99.4_{0.2}$** | **$84.3_{0.4}$** | **$80.1_{0.7}$** | $80.9_{0.7}$ | **$82.3_{1.4}$** |

### 6.1. Experimental Setup

We evaluate on 7 molecular property prediction benchmarks from MoleculeNet (Wu et al., 2018), comparing against SOTA methods spanning message-passing networks, pre-training approaches, geometry-aware models, and transformers (Table 1). Details are provided in Appendix C.

### 6.2. Main Results

**Geometry-sensitive improvements.** SPD-Sheaf model achieves the best performance on 6 out of 7 benchmarks and ranks second on the remaining one (Table 1). Substantial improvements appear on geometry-sensitive tasks. On BBBP, we observe a 4.0% absolute improvement (77.4% vs. 73.4%), suggesting that shape-dependent properties benefit from second-order geometric information captured by SPD sheaves. On ClinTox , we achieve 99.4% ROC-AUC, improving upon the geometry-aware baseline EGNN by 0.8%. The consistent gains on BACE (+1.7%) and SIDER (+1.1%) further corroborate that protein-ligand binding and drug side-effect prediction are geometry-sensitive tasks where second-order representations provide meaningful benefits.

**Comparison with pre-training methods.** On HIV, SPD-Sheaf ranks second (80.9% vs. 81.2%). SMPT relies on pre-training on millions of molecules, whereas SPD-Sheaf is trained from scratch. Achieving competitive performance without pre-training suggests that the inductive bias of SPD geometry can partially offset the lack of external data.

### 6.3. Ablation Studies: Disentangling Contributions

Table 2 systematically isolates the contributions of SPD geometry and sheaf structure.

**SPD geometry matters.** Comparing SPD-Sheaf with Euclidean SheafNN, we observe consistent improvements across all datasets (+3.3% on BBBP, +1.5% on BACE, +7.0% on MUV), consistent with Theorem 4.15: SPD-

*Table 2.* Disentangling SPD and sheaf contributions. All variants use the same semantic stream and fusion mechanism; only the geometric module differs. ROC-AUC (%) with standard deviations.[†] Averaged over 3 runs due to computational cost.

| Geometric Module | BBBP | BACE | ClinTox | SIDER | Tox21 | HIV | MUV |
|---|---|---|---|---|---|---|---|
| GCN (baseline) | $73.9_{1.9}$ | $87.0_{1.0}$ | $98.4_{0.2}$ | $82.6_{0.1}$ | $78.7_{0.3}$ | $80.4_{1.0}$ | $75.7_{3.0}$ |
| *Effect of sheaf structure (Euclidean stalks):* | | | | | | | |
| Euclidean SheafNN (Barbero et al., 2022) | $74.1_{1.8}$ | $87.5_{0.9}$ | $99.0_{0.4}$ | $83.9_{0.2}$ | $79.4_{0.6}$ | $80.9_{0.7}$ | $75.3^{†}_{1.1}$ |
| *Effect of SPD geometry (no sheaf structure):* | | | | | | | |
| SPD4GNN (Zhao et al., 2023) | $71.9_{2.8}$ | $87.5_{0.9}$ | $98.7_{0.4}$ | $83.9_{0.3}$ | $79.3_{0.8}$ | $79.6_{1.2}$ | $76.2_{4.6}$ |
| HyperbolicGCN (Chami et al., 2019) | $73.3_{2.3}$ | $87.3_{0.8}$ | $98.6_{0.6}$ | $84.1_{0.2}$ | $79.3_{0.9}$ | $80.7_{1.2}$ | $79.2_{2.6}$ |
| *Combined: SPD geometry + sheaf structure:* | | | | | | | |
| **SPD Sheaf (Ours)** | $\mathbf{77.4_{2.7}}$ | $\mathbf{89.0_{1.4}}$ | $\mathbf{99.4_{0.2}}$ | $\mathbf{84.3_{0.4}}$ | $\mathbf{80.1_{0.7}}$ | $\mathbf{80.9_{0.7}}$ | $\mathbf{82.3_{1.4}}$ |

*Table 3.* Ablation on architectural components. $\Delta$ indicates change from the full model.

| Variant | BBBP | BACE | ClinTox | SIDER | Tox21 | HIV | MUV |
|---|---|---|---|---|---|---|---|
| **Full model** | $77.4_{2.7}$ | $89.0_{1.4}$ | $99.4_{0.2}$ | $84.3_{0.4}$ | $80.1_{0.7}$ | $80.9_{0.7}$ | $82.3_{1.4}$ |
| w/o Semantic stream | $71.8_{2.0}$ | $71.6_{4.1}$ | $97.9_{0.6}$ | $83.2_{0.5}$ | $68.0_{0.7}$ | $66.2_{1.5}$ | $58.4_{0.9}$ |
| $\Delta$ | $-5.6$ | $-17.4$ | $-1.5$ | $-1.1$ | $-12.1$ | $-14.7$ | $-23.9$ |
| w/o Geometric stream | $73.3_{3.2}$ | $82.1_{1.4}$ | $98.1_{0.7}$ | $84.3_{0.2}$ | $76.3_{0.3}$ | $74.8_{1.4}$ | $69.8_{0.5}$ |
| $\Delta$ | $-4.1$ | $-6.9$ | $-1.3$ | $0.0$ | $-3.8$ | $-6.1$ | $-12.5$ |
| w/o Cross-modal interaction | $75.2_{0.6}$ | $86.9_{1.3}$ | $99.4_{0.3}$ | $84.3_{0.3}$ | $79.8_{0.8}$ | $80.3_{0.6}$ | $80.8_{2.0}$ |
| $\Delta$ | $-2.2$ | $-2.1$ | $0.0$ | $0.0$ | $-0.3$ | $-0.6$ | $-1.5$ |

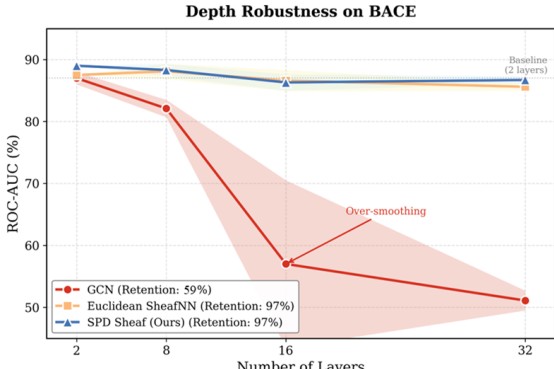

*Figure 3.* Both sheaf methods resist over-smoothing, with SPD-Sheaf achieving the best performance across most depths.

valued stalks provide enhanced representational capacity.

**Sheaf structure matters.** SPD Sheaf also outperforms both standard GNNs (+4.6% over GCN on BBBP) and methods using SPD geometry without sheaf structure (+6.1% over SPD4GNN, +3.1% over HyperbolicGCN on MUV), confirming that edge-specific transformations provide complementary benefits (Remark 4.9). We further verify this mechanistically via a controlled graph perturbation analysis (Appendix I).

**Combined benefit.** Neither component alone matches the full model, supporting our thesis that SPD geometry and sheaf structure address distinct challenges: the former captures richer second-order information, while the latter enables principled edge-specific propagation.

### 6.4. Architectural Ablations

Table 3 examines the dual-stream design. Both streams are essential: removing the semantic stream causes severe degradation (up to 23.9% on MUV), confirming geometry alone is insufficient. Removing the geometric stream also degrades performance, particularly on BACE ($-6.9\%$) and MUV ($-12.5\%$), demonstrating the importance of second-order geometric representations. Cross-modal interaction improves most datasets, validating the design choice of allowing geometry to modulate semantic learning. Note that the dual-stream design is tailored for molecular data, where chemical semantics complement geometric structure. In Appendix E, the SPD-Sheaf stream alone achieves competitive performance on EEG classification, confirming the standalone effectiveness of our geometric framework.

### 6.5. Analysis

**Depth Robustness.** A key advantage of sheaf methods is resistance to over-smoothing. Figure 3 evaluates this on

BACE with varying network depths. GCN exhibits severe performance degradation: ROC-AUC drops from 87.0% at 2 layers to 51.1% at 32 layers, a decline of over 35% confirming that standard message passing suffers from representation collapse at depth. In contrast, both sheaf methods maintain remarkable stability. At 32 layers, SPD-Sheaf retains 97% of its peak performance (86.7% vs. 89.0%) while GCN retains only 59%. We further quantify this behavior using normalized Dirichlet energy and Mean Average Distance across BACE, Cora, and Texas at depths up to 128 (Appendix H).

Two observations merit discussion. First, both sheaf formulations effectively resist over-smoothing, confirming that edge-specific transformations preserve node distinctiveness (Remark 4.9). Second, SPD outperforms Euclidean sheaf across most depths, and notably achieves its best performance at just 2 layers (89.0%), suggesting that SPD geometry enables more efficient information propagation and extracts richer geometric representations in fewer layers.

**From boundary to interior: Second-order emergence.**

A key finding of our work is that SPD sheaf convolution automatically lifts first-order directional inputs into second-order geometric representations (Figure 1, top). We track the *effective rank* of node-level SPD matrices across layers, defined as $\mathrm{erank}(X) = \exp(H(\boldsymbol{\lambda}))$, where $H(\boldsymbol{\lambda}) = -\sum_i \bar{\lambda}_i \log \bar{\lambda}_i$ is the normalized eigenvalue entropy. For a $3 \times 3$ SPD matrix with eigenvalues $(\lambda, \varepsilon, \varepsilon)$ where $\varepsilon \ll \lambda$, $\mathrm{erank} \approx 1$; for equal eigenvalues, $\mathrm{erank} = 3$.

Table 4 confirms this phenomenon across all datasets. Initial SPD matrices $\hat{u}_v \hat{u}_v^\top + \varepsilon I$ have effective rank $\approx 1.00$, encoding essentially first-order directional information. After sheaf propagation, effective rank increases substantially (1.72–2.28), with the second eigenvalue $\lambda_2$ growing from near-zero to values exceeding 1.1.

This observation admits a geometric interpretation. The initialization $\hat{u}_v \hat{u}_v^\top + \varepsilon I$ lies near the *boundary* of the SPD manifold—nearly rank-deficient matrices occupying a lower-dimensional submanifold. Through sheaf convolution

*Table 4.* SPD sheaf convolution transforms effectively rank-1 initializations into higher rank matrices across all datasets.

| Dataset | Layers | Initial Erank | Final Erank | Δ Erank | $\lambda_2$: Init→Final |
|---------|--------|---------------|-------------|---------|-------------------------|
| BACE | 2 | 1.00 | 2.28 | +1.28 | 0.00 → 1.16 |
| BBBP | 2 | 1.00 | 2.16 | +1.16 | 0.00 → 1.15 |
| SIDER | 2 | 1.00 | 2.22 | +1.22 | 0.00 → 1.14 |
| ClinTox | 5 | 1.00 | 1.83 | +0.83 | 0.00 → 1.32 |
| HIV | 5 | 1.00 | 1.87 | +0.87 | 0.00 → 1.42 |
| Tox21 | 7 | 1.00 | 1.75 | +0.75 | 0.00 → 1.46 |
| MUV | 7 | 1.00 | 1.72 | +0.72 | 0.00 → 1.55 |

via the Laplacian, these boundary representations evolve into the *interior* of $\mathrm{SPD}_n$, becoming full-rank matrices that span the entire manifold. Intuitively, the sheaf Laplacian aggregates directional information from neighbors; when these directions are non-collinear, the aggregation naturally produces matrices with multiple significant eigenvalues, encoding local geometric structure.

This mechanism explains the empirical gains on geometry-sensitive tasks: by evolving from boundary (erank-1, first-order) to interior (full-rank, second-order), SPD sheaf representations capture richer structural information including bond angle distributions, ring planarity, local anisotropy, that vector-valued methods fundamentally cannot express.

# 7. Conclusions

We developed the first sheaf neural network on SPD manifolds. By exploiting the Lie group structure of $\mathrm{SPD}_n$, we constructed sheaf operators that respect non-Euclidean geometry while preserving classical algebraic structure. Our theoretical analysis establishes that SPD sheaves strictly generalize Euclidean sheaves (Theorem 4.15).

Empirically, our dual-stream architecture achieves SOTA on 6 out of 7 MoleculeNet benchmarks. We observe that sheaf convolution transforms erank-1 initializations into full-rank representations, encoding local geometric structure. Ablations confirm complementary benefits from SPD geometry and sheaf structure, while 97% performance retention at 32 layers demonstrates depth robustness.

## 7.1. Limitation & Future Work

**Sign ambiguity of SPD embeddings.** One limitation of the SPD embedding $\Phi_\varepsilon(v) = vv^\top + \varepsilon I$ is that it removes sign information, since $vv^\top = (-v)(-v)^\top$. Thus, when the data carries directional information and $v$ and $-v$ correspond to physically distinct states, this sign loss may be undesirable. However, whether this ambiguity is a limitation depends on the task. For sign-insensitive targets, the same ambiguity can instead act as a useful inductive bias. As discussed in Remark 4.14, SPD sheaves naturally quotient out the $\mathbb{Z}_2$ sign ambiguity, eliminating the line-bundle frustrations that can obstruct global sections in Euclidean sheaves and thereby preserving geometric information that Euclidean sheaves may lose due to sign conflicts. Theorem 4.15 for-

malizes this advantage: the global-section space of the SPD sheaf strictly contains that of the corresponding Euclidean sheaf.

This is precisely the regime considered in our molecular property prediction experiments. Molecular properties are determined primarily by relative atomic geometry and intrinsic second-order structure, not by an arbitrary absolute sign of an embedding vector. Quotienting out the sign degree of freedom therefore removes a confounding factor that is irrelevant to the target properties, allowing the model to focus on sign-invariant geometric structure. This interpretation is consistent with our empirical results, where SPD-Sheaf achieves state-of-the-art performance on six out of seven MoleculeNet benchmarks.

**From invariance to equivariance.** A second limitation concerns the symmetry properties of the current architecture: SPDSheaf is invariant by design. While this suffices for scalar molecular property prediction, where targets are intrinsic to the molecule, tasks such as force field prediction, stress tensor estimation, and molecular dynamics demand equivariant outputs, a regime where SOTA methods (e.g., MACE, NequIP) build equivariance directly into the architecture. Our framework, however, admits a natural path to this extension. Since the Lie group operations $\log$ and $\exp$ commute with orthogonal conjugation, the only requirement is that restriction maps transform equivariantly, i.e., $M_{ve} \mapsto R M_{ve} R^\top$ under a global rotation $R$. One concrete construction is to decompose $X_v = Q_v \Lambda_v Q_v^\top$ and define $M_{ve} = Q_v \hat{M} Q_v^\top$, where $\hat{M} \in O(n)$ is a learnable orthogonal matrix predicted from invariant features, yielding equivariance without modifying the core SPD sheaf convolution. We regard the development of a natively equivariant SPD-Sheaf, and its evaluation on geometry-sensitive equivariant tasks, as an important direction for future work.

# Acknowledgements

This work was supported in part by the Singapore Ministry of Education Academic Research Fund Tier 1 (RG16/23) and Tier 2 (MOE-T2EP20125-0004, MOE-T2EP50223-0036).

It was further supported by the National Key R&D Program of China (Grant No. 2024YFA1013202), the National Natural Science Foundation of China (Grant No. 11221091), the Fundamental Research Funds for the Central Universities, and the Nankai Zhide Foundation.

A.W. thanks the Hector Fellow Academy for support.

# Impact Statement

This paper presents work whose goal is to advance the field of Machine Learning. There are many potential societal consequences of our work, none of which we feel must be specifically highlighted here.

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

# Appendix

# A. Proofs

### A.1. Geodesic completeness of Lemma 3.2

A Riemannian manifold $(\mathcal{M}, g)$ is said to be *(geodesically) complete* if every maximal geodesic $\gamma : I \to \mathcal{M}$ is defined on the entire real line, i.e., $I = (-\infty, \infty)$. Equivalently, $M$ is (geodesically) complete if for all points $p \in M$, the exponential map at $p$ is defined on $T_p M$, the entire tangent space at $p$.

### A.2. Proof of Proposition 4.2 (Restriction Maps are Isometries)

*Proof.* For any $X, Y \in \mathrm{SPD}_n$ and $M \in O(n)$:

$$
\begin{aligned}
d_{\mathrm{AI}}^2(MXM^\top, MYM^\top) &= \left\| \log\left( (MXM^\top)^{-1/2}(MYM^\top)(MXM^\top)^{-1/2} \right) \right\|_F^2 \\
&= \left\| \log\left( M^{-\top}X^{-1/2}M^{-1} \cdot MYM^\top \cdot M^{-\top}X^{-1/2}M^{-1} \right) \right\|_F^2 \\
&= \left\| \log\left( MX^{-1/2}YX^{-1/2}M^\top \right) \right\|_F^2 \\
&= \left\| M \log(X^{-1/2}YX^{-1/2})M^\top \right\|_F^2 \\
&= \left\| \log(X^{-1/2}YX^{-1/2}) \right\|_F^2 \\
&= d_{\mathrm{AI}}^2(X, Y),
\end{aligned}
$$

$$
\begin{aligned}
d_{\mathrm{LE}}(MXM^\top, MYM^\top) &= \left\| \log(MXM^\top) - \log(MYM^\top) \right\|_F \\
&= \left\| M(\log X - \log Y)M^\top \right\|_F \\
&= \left\| \log X - \log Y \right\|_F \\
&= d_{\mathrm{LE}}(X, Y).
\end{aligned}
$$

where we used $M^\top = M^{-1}$ (orthogonality), the identity $\log(MAM^\top) = M(\log A)M^\top$ for symmetric $A$, and the orthogonal invariance of the Frobenius norm: $\|MAM^\top\|_F = \|A\|_F$.

$\square$

### A.3. Proof of Proposition 4.4 (Linearity of Coboundary)

*Proof.* We first establish that restriction maps are group homomorphisms with respect to the Lie group operation $\odot$, i.e. For any $X, Y \in \mathrm{SPD}_n$ and $M \in O(n)$: $\mathcal{F}_{v \to e}(X \odot Y) = \mathcal{F}_{v \to e}(X) \odot \mathcal{F}_{v \to e}(Y)$.

$$
\begin{aligned}
\mathcal{F}_{v \to e}(X \odot Y) &= M \exp(\log X + \log Y)M^\top \\
&= M \left( \sum_{k=0}^\infty \frac{(\log X + \log Y)^k}{k!} \right) M^\top \\
&= \sum_{k=0}^\infty \frac{M(\log X + \log Y)^k M^\top}{k!} \\
&= \sum_{k=0}^\infty \frac{(M(\log X + \log Y)M^\top)^k}{k!} \\
&= \exp(M(\log X + \log Y)M^\top) \\
&= \exp(M \log X M^\top + M \log Y M^\top).
\end{aligned}
$$

On the other hand:

$$\mathcal{F}_{v\to e}(X) \odot \mathcal{F}_{v\to e}(Y) = \exp(\log(MXM^\top) + \log(MYM^\top))$$
$$= \exp(M \log X M^\top + M \log Y M^\top).$$

Thus, $\mathcal{F}_{v\to e}(X \odot Y) = \mathcal{F}_{v\to e}(X) \odot \mathcal{F}_{v\to e}(Y)$. We now use this fact to verify the linearity of $\delta$ under $\odot$.

$$
\begin{aligned}
(\delta(\sigma \odot \tau))_e &= \exp\left(\log(\mathcal{F}_{v\to e}(\sigma_v \odot \tau_v)) - \log(\mathcal{F}_{u\to e}(\sigma_u \odot \tau_u))\right) \\
&= \exp\left(\log(\mathcal{F}_{v\to e}(\sigma_v) \odot \mathcal{F}_{v\to e}(\tau_v)) - \log(\mathcal{F}_{u\to e}(\sigma_u) \odot \mathcal{F}_{u\to e}(\tau_u))\right) \\
&= \exp\Big(\log(\mathcal{F}_{v\to e}(\sigma_v)) + \log(\mathcal{F}_{v\to e}(\tau_v)) \\
&\qquad - \log(\mathcal{F}_{u\to e}(\sigma_u)) - \log(\mathcal{F}_{u\to e}(\tau_u))\Big) \\
&= \exp\Big([\log(\mathcal{F}_{v\to e}(\sigma_v)) - \log(\mathcal{F}_{u\to e}(\sigma_u))] \\
&\qquad + [\log(\mathcal{F}_{v\to e}(\tau_v)) - \log(\mathcal{F}_{u\to e}(\tau_u))]\Big) \\
&= (\delta\sigma)_e \odot (\delta\tau)_e.
\end{aligned}
$$

$\square$

### A.4. Proof of Proposition 4.6 (Adjoint Operator)

*Proof.* **Overview.** The goal of this appendix is to construct an adjoint coboundary operator $\delta^\top : C^1 \to C^0$ satisfying the Green identity

$$\langle \delta\sigma, \tau \rangle_{C^1} = \langle \sigma, \delta^\top \tau \rangle_{C^0}.$$

Since the space $\mathrm{SPD}_n$ is not a vector space, it does not admit a canonical bilinear pairing on its points; consequently, the adjoint operator is not intrinsic for $\mathrm{SPD}_n$-valued cochains and must be defined relative to a chosen pairing.

We therefore proceed in three steps: (i) we define a bilinear pairing on $\mathrm{SPD}_n$ by transporting a fixed pairing on the tangent space $T_{I_n}\mathrm{SPD}_n$ via the global logarithm map; (ii) under this pairing, we construct an adjoint operator $\delta^\top$ as the (unique) operator satisfying the Green identity (A.4); and (iii) we derive the explicit form of $\delta^\top$.

**Log-induced pairing and the meaning of $g_{I_n}$.** Let $(M, g)$ be a complete Riemannian manifold and fix a base point $p \in M$. Assume that the Riemannian exponential map at $p$,

$$\exp_p : T_p M \to M,$$

is a global diffeomorphism, so that its inverse $\log_p : M \to T_p M$ is well-defined on all of $M$.

Denote by $g_p$ the inner product on the tangent space $T_p M$ induced by the Riemannian metric $g$ at the point $p$. Using the inner product $g_p$ on $T_p M$, we define a *log-induced pairing* between points $x, y \in M$ by

$$\langle x, y \rangle_p := g_p\big(\log_p(x), \log_p(y)\big).$$

Moreover, the logarithm map induces a base-point dependent "addition" on $M$:

$$x \odot_p y := \exp_p\big(\log_p(x) + \log_p(y)\big),$$

under which the pairing is bilinear in the sense that $\langle x \odot_p x', y \rangle_p = \langle x, y \rangle_p + \langle x', y \rangle_p$ and similarly for the second argument.

Importantly, both $\langle \cdot, \cdot \rangle_p$ and $\odot_p$ depend on the choice of the base point $p$. We therefore regard it as a base-point-dependent structure, rather than the standard pointwise Riemannian inner products $g_x$ defined on the tangent spaces $T_x M$.

In our setting $M = \mathrm{SPD}_n$, we take $p = I_n$ so that the resulting pairing is compatible with the matrix logarithm used in Lie group computations. This explains the notation $g_{I_n}$ used in the main text: it denotes the fixed bilinear pairing on $T_{I_n}\mathrm{SPD}_n$ transported to $\mathrm{SPD}_n$ via the global logarithm map.

Since the construction of the pairing depends on a choice of basepoint, it is natural to ask how the pairing varies when the basepoint is perturbed. The following proposition characterizes this dependence precisely.

**Proposition A.1.** *For any $p \in U$ and $V \in T_pM$, the covariant derivative of the vector field $X(p) = \log_p(x)$ in direction $V$ is given by*

$$\nabla_V X = J'(0),$$

*where $J(t)$ is the unique Jacobi field along the geodesic $\sigma(t) = \exp_p(tX(p))$ satisfying the boundary conditions*

$$J(0) = V, \qquad J(1) = 0.$$

*Proof.* Let $\gamma : (-\epsilon, \epsilon) \to U$ be a smooth curve such that $\gamma(0) = p$ and $\gamma'(0) = V$. Consider the smooth variation of geodesics $\alpha : (-\epsilon, \epsilon) \times [0, 1] \to M$ defined by

$$\alpha(s, t) = \exp_{\gamma(s)}\big(t \, X(\gamma(s))\big).$$

For each fixed $s$, the curve $t \mapsto \alpha(s, t)$ is the geodesic connecting $\gamma(s)$ to $x$, hence the variation has boundary values

$$\alpha(s, 0) = \gamma(s), \qquad \alpha(s, 1) = x.$$

By construction, the initial velocity of this geodesic is

$$X(\gamma(s)) = \frac{\partial \alpha}{\partial t}(s, 0).$$

Therefore, the covariant derivative of $X$ along $V = \gamma'(0)$ can be written as

$$\nabla_V X = \frac{D}{ds} X(\gamma(s)) \Big|_{s=0} = \frac{D}{ds} \frac{\partial \alpha}{\partial t}(s, t) \Big|_{s=0,\, t=0}.$$

Since the Levi–Civita connection is torsion-free, the covariant derivatives commute for this two-parameter variation:

$$\frac{D}{ds} \frac{\partial \alpha}{\partial t} = \frac{D}{dt} \frac{\partial \alpha}{\partial s}.$$

Consequently,

$$\nabla_V X = \frac{D}{dt} \frac{\partial \alpha}{\partial s}(0, t) \Big|_{t=0}.$$

Define the variational vector field along $\sigma(t) := \alpha(0, t)$ by

$$J(t) := \frac{\partial \alpha}{\partial s}(0, t).$$

Because $\alpha$ is a variation through geodesics, $J$ is a Jacobi field and satisfies the Jacobi equation

$$J''(t) + R\big(J(t), \dot{\sigma}(t)\big)\dot{\sigma}(t) = 0.$$

Substituting the definition of $J$ yields

$$\nabla_V X = \frac{D}{dt} J(t) \Big|_{t=0} = J'(0).$$

Finally, the boundary conditions follow from the variation boundaries:

$$J(0) = \frac{\partial \alpha}{\partial s}(0, 0) = \gamma'(0) = V, \qquad J(1) = \frac{\partial \alpha}{\partial s}(0, 1) = 0,$$

where $J(1) = 0$ holds because $\alpha(s, 1) = x$ is constant in $s$. $\square$

In summary, the first-order change of the "pair" is explicitly encoded by the Jacobi field $J$ by viewing $\langle x, y \rangle_p$ as function of $p \in M$,

$$V \langle x, y \rangle_p = V g_p \left( X(p), Y(p) \right) = \left( g \left( \nabla_V X, Y \right) + g \left( X, \nabla_V Y \right) \right)(p)$$

Hence it depends on the geometric information carried by the metric $g$.

**Adjoint and Green identity.** We begin by establishing the linearity of the restriction maps in logarithmic coordinates. Let the restriction map be given by orthogonal congruence

$$\mathcal{F}_{v\to e}(X) = M_{ve} X M_{ve}^\top, \qquad M_{ve} \in O(n).$$

Since orthogonal conjugation preserves the spectral decomposition, the matrix logarithm commutes with $\mathcal{F}_{v\to e}$, yielding

$$\log(\mathcal{F}_{v\to e}(X)) = M_{ve}(\log X)M_{ve}^\top =: A_{ve}(\log X),$$

where $A_{ve} : \mathrm{Sym}(n) \to \mathrm{Sym}(n)$ is a linear operator.

And this implies that the coboundary operator is linear in logarithmic coordinates. The coboundary operator is defined edgewise by

$$(\delta\sigma)_e = \exp\Big( \log(\mathcal{F}_{v\to e}(\sigma_v)) - \log(\mathcal{F}_{u\to e}(\sigma_u)) \Big).$$

Taking the logarithm gives

$$\log(\delta\sigma)_e = A_{v\to e} \log(\sigma_v) - A_{u\to e} \log(\sigma_u).$$

Collecting all vertex-wise logarithms into a vector $z := \log\sigma \in \bigoplus_{v\in V} \mathrm{Sym}(n)$, there exists a linear operator

$$B : \bigoplus_{v\in V} \mathrm{Sym}(n) \to \bigoplus_{e\in E} \mathrm{Sym}(n)$$

such that

$$\log(\delta\sigma) = B \log\sigma.$$

With respect to the fixed bilinear pairing on the tangent space $T_{I_n}\mathrm{SPD}_n$, the linear operator $B$ admits an adjoint $B^\top$ satisfying

$$\langle B \log\sigma, \log\tau\rangle = \langle \log\sigma, B^\top \log\tau\rangle,$$

where $B^\top$ denotes the adjoint of $B$ with respect to the chosen pairing on $T_{I_n}\mathrm{SPD}_n$. This motivates defining the adjoint coboundary operator $\delta^\top : C^1 \to C^0$ via

$$\log(\delta^\top\tau) := B^\top \log\tau.$$

Consequently, we obtain

$$\langle \log(\delta\sigma), \log\tau\rangle = \langle \log\sigma, \log(\delta^\top\tau)\rangle,$$

which yields the Green identity

$$\langle \delta\sigma, \tau\rangle_{C^1} = \langle \sigma, \delta^\top\tau\rangle_{C^0}.$$

**Derive the explicit form of $\delta^\top$.** Now, we derive the adjoint by requiring $\langle \delta x, y\rangle = \langle x, \delta^\top y\rangle$ for all 0-cochains $x$ and 1-cochains $y$. We denote in this part that $i$ for index of vertices and $j$ for index of edges. For the case of vertex $i$ belongs to edge $j$ (denoted by $i \to j$), denote

$$I(i, j) = \begin{matrix} 0 & i \text{ is the start of arrow} \\ 1 & i \text{ is the end of arrow} \end{matrix}$$

We have

$$\big\langle \delta(x_1, \ldots, x_n), (y_1, \ldots, y_m)\big\rangle = \big\langle (x_1, \ldots, x_n), \delta^\top(y_1, \ldots, y_m)\big\rangle.$$

$$\Rightarrow \sum_i \langle x_i, \delta^\top(y)_i\rangle = \sum_j \langle y_j, \delta(x)_j\rangle.$$

Let $x = (I_d, \ldots, x_i, \ldots, I_d)$, then

$$\delta(x)_j = \begin{cases} \mathcal{F}_{i\to j}x_i = M_{ij}x_i M_{ij}^\top, & I(i,j) = 0, \\ (\mathcal{F}_{i\to j}x_i)^{-1} = M_{ij}x_i^{-1}M_{ij}^\top, & I(i,j) = 1. \end{cases}$$

Thus

$$\langle x_i, \delta^\top(y)_i\rangle = \sum_j \langle y_j, \delta(x)_j\rangle = \sum_{i\to j} \langle y_j, M_{ij} x_i^{(-1)^{I(i,j)}} M_{ij}^\top\rangle.$$

Equivalently,

$$g_{I_n}\big(\log x_i, \ \log \delta^\top(y)_i\big) = \sum_{i\to j} g_{I_n}\big(\log y_j, \ \log \delta(x)_j\big).$$

Let $\{e_i\}$ be an orthonormal frame at $T_{I_d}\mathrm{SPD}_n$, and write $x_i = \exp(e_k)$.
Then

$$
\begin{aligned}
\log \delta^\top(y)_i &= \sum_k g_{I_n}\big(\log\delta^\top(y)_i, \ e_k\big)e_k.\\
&= \sum_k\sum_{i\to j} g_{I_n}\big(\log y_j, \ \log\delta(x)_j\big)e_k\\
&= \sum_k\sum_{i\to j} g_{I_n}\Big(\log y_j, \ M_{ij}(-1)^{I(i,j)}e_k M_{ij}^\top\Big)e_k\\
&= \sum_{i\to j}\sum_k (-1)^{I(i,j)} g_{I_n}\big(\log y_j, \ M_{ij}e_k M_{ij}^\top\big)e_k.\\
&= \sum_{i\to j}\sum_k (-1)^{I(i,j)} M_{ij}^\top\Big[g_{I_n}\big(\log y_j, \ M_{ij}e_k M_{ij}^\top\big)M_{ij}e_k M_{ij}^\top\Big]M_{ij}.\\
&= \sum_{i\to j} M_{ij}^\top\left(\sum_k(-1)^{I(i,j)} g_{I_n}\big(\log y_j, \ M_{ij}e_k M_{ij}^\top\big)M_{ij}e_k M_{ij}^\top\right)M_{ij}.\\
&= \sum_{i\to j} M_{ij}^\top\Big[(-1)^{I(i,j)}\log y_j\Big]M_{ij}.
\end{aligned}
$$

Finally,

$$\delta^\top(y)_i = \exp\sum_{i\to j}(-1)^{I(i,j)} M_{ij}^\top \log y_j M_{ij}.$$

$\square$

### A.5. Proof of Proposition 4.8 (Hodge-type Decomposition for SPD-valued Sheaves)

*Proof.* Since we have already clarified that the pairing $\langle\cdot,\cdot\rangle$ defined on $\mathrm{SPD}_n\times\mathrm{SPD}_n$ on cochains is compatible with the group structure on $\mathrm{SPD}_n$ and is positive definite in the sense that

$$\langle x,x\rangle \geq 0, \qquad \langle x,x\rangle = 0 \iff x = e$$

where $e$ denotes the identity element of $\mathrm{SPD}_n$. We claim that the Laplacian $\mathbf{L}$ satisfies a property analogous to the Hodge decomposition in sheaf cohomology.

More precisely, we consider the following complex:

$$0 \longrightarrow C^0(\mathcal{F}) \xrightarrow{\delta} C^1(\mathcal{F}) \longrightarrow 0.$$

One can define the associated cohomology groups as

$$H^0 := \ker\delta, \qquad H^1 := C^1(\mathcal{F})/\operatorname{Im}\delta.$$

We also define the Laplacian operator as

$$\mathbf{L} : C^0(\mathcal{F}) \longrightarrow C^0(\mathcal{F}).$$

The inclusion $\ker\delta \subseteq \ker\mathbf{L}$ is immediate: if $\delta x = e$, then $\mathbf{L}x = \delta^\top(\delta x) = \delta^\top(e) = e$.

Conversely, let $x \in C^0(\mathcal{F})$ satisfy $\mathbf{L}x = e$. Taking the pairing with $x$ and using the defining property of $\delta^\top$, we obtain

$$\langle \delta x, \delta x \rangle_{C^1} = \langle \delta^\top \delta x, x \rangle_{C^0} = \langle \mathbf{L}x, x \rangle_{C^0} = 0$$

By positive definiteness of the pairing on $C^1$, this implies $\delta x = e$, i.e. $x \in \ker \delta$ and therefore $\ker \mathbf{L} \subseteq \ker \delta$.

This result provides a geometric interpretation: the limit of the heat flow on an SPD-valued sheaf coincides with its coboundary-free component, analogous to the classical interpretation of the Laplacian on Riemannian manifolds. $\qquad\square$

### A.6. Proof of Proposition 4.11 (Restriction Maps as Pullbacks)

*Proof.* Let $\Sigma \in \mathrm{SPD}_n$ define an inner product on $\mathbb{R}^n$ by $\langle x, y \rangle_\Sigma = x^\top \Sigma y$. The pullback of this inner product under a linear map $F : \mathbb{R}^n \to \mathbb{R}^n$ is defined by:

$$(F^*\Sigma)(x, y) = \Sigma(F_* x, F_* y) = \Sigma(Fx, Fy) = (Fx)^\top \Sigma(Fy) = x^\top (F^\top \Sigma F) y.$$

For the Euclidean restriction map $\mathcal{F}_{v \to e} = M_{ve}$ where $M_{ve} \in O(n)$:

$$(\mathcal{F}_{v \to e})^* \Sigma(x, y) = x^\top (M_{ve}^\top \Sigma M_{ve}) y = (M_{ve}x)^\top \Sigma(M_{ve}y) = \Sigma(M_{ve}x, M_{ve}y)$$

However, the SPD restriction map is defined as $\mathcal{G}_{v \to e}(\Sigma) = M_{ve} \Sigma M_{ve}^\top$. Let us verify the pullback relationship more carefully:

$$\begin{aligned}
\mathcal{G}_{v \to e}(\Sigma)(x, y) &= x^\top (M_{ve} \Sigma M_{ve}^\top) y \\
&= (M_{ve}^\top x)^\top \Sigma(M_{ve}^\top y) \\
&= \Sigma(M_{ve}^\top x, M_{ve}^\top y).
\end{aligned}$$

This shows that $\mathcal{G}_{v \to e} = (\mathcal{F}_{v \to e}^T)^*$, which for orthogonal $\mathcal{F}_{v \to e}$ gives the stated pullback relationship. $\qquad\square$

### A.7. Proof of Proposition 4.13 (Kernel Correspondence)

*Proof.* Let $\mathcal{F}$ be a Euclidean sheaf and $\mathcal{G}$ an SPD sheaf with matched restriction maps. For every edge $e = (i, j)$, we show the equivalence step by step.

($\Rightarrow$) Suppose $\delta_{\mathcal{F}}(\mathbf{x})_e = 0$. Then:

$$\begin{aligned}
\mathcal{F}_{i \to e} x_i - \mathcal{F}_{j \to e} x_j &= 0 \\
M_{ie} x_i &= M_{je} x_j.
\end{aligned}$$

Now consider the SPD cochain $\Phi(\mathbf{x})$ with $\Phi(x_i) = x_i x_i^\top + \varepsilon I$:

$$\begin{aligned}
\mathcal{G}_{i \to e}(\Phi(x_i)) &= M_{ie}(x_i x_i^\top + \varepsilon I) M_{ie}^\top \\
&= M_{ie} x_i (M_{ie} x_i)^\top + \varepsilon I \\
&= M_{je} x_j (M_{je} x_j)^\top + \varepsilon I \quad (\text{using } M_{ie} x_i = M_{je} x_j) \\
&= M_{je}(x_j x_j^\top + \varepsilon I) M_{je}^\top \\
&= \mathcal{G}_{j \to e}(\Phi(x_j)).
\end{aligned}$$

Therefore:

$$\delta_{\mathcal{G}}(\Phi(\mathbf{x}))_e = \exp(\log \mathcal{G}_{i \to e}(\Phi(x_i)) - \log \mathcal{G}_{j \to e}(\Phi(x_j))) = \exp(0) = I_n.$$

($\Leftarrow$) Suppose $\delta_{\mathcal{G}}(\Phi(\mathbf{x}))_e = I_n$. Then:

$$\begin{aligned}
\exp(\log \mathcal{G}_{i \to e}(\Phi(x_i)) - \log \mathcal{G}_{j \to e}(\Phi(x_j))) &= I_n \\
\log \mathcal{G}_{i \to e}(\Phi(x_i)) &= \log \mathcal{G}_{j \to e}(\Phi(x_j)) \\
\mathcal{G}_{i \to e}(\Phi(x_i)) &= \mathcal{G}_{j \to e}(\Phi(x_j)).
\end{aligned}$$

This means:
$$M_{ie}(x_i x_i^\top + \varepsilon I)M_{ie}^\top = M_{je}(x_j x_j^\top + \varepsilon I)M_{je}^\top.$$

Let $y_i = M_{ie}x_i$ and $y_j = M_{je}x_j$. Then:
$$y_i y_i^\top + \varepsilon I = y_j y_j^\top + \varepsilon I \implies y_i y_i^\top = y_j y_j^\top.$$

For rank-1 matrices, $y_i y_i^\top = y_j y_j^\top$ implies $y_i = \pm y_j$. This induces the feedback result $\delta_\mathcal{F}(|x|) = 0$. □

### A.8. Proof of Theorem 4.15 (Strict Extension)

*Proof.* We prove the theorem in two parts: (1) $\Phi(\ker \delta_\mathcal{F}) \subseteq \ker \delta_\mathcal{G}$, and (2) the inclusion is strict.

**Part 1: Inclusion.** Let $\mathcal{F}$ be a Euclidean sheaf and $\mathcal{G}$ an SPD sheaf with matched restriction maps. By Proposition 4.13, for any $\mathbf{x} \in \ker \delta_\mathcal{F}$, we have
$$\delta_\mathcal{F}\mathbf{x} = 0 \implies \delta_\mathcal{G}(\Phi(\mathbf{x}))_e = I_n \quad \text{for all edges } e.$$

By definition,
$$\ker \delta_\mathcal{G} = \{\sigma \mid \delta_\mathcal{G}(\sigma)_e = I_n \text{ for all } e\},$$

and therefore $\Phi(\mathbf{x}) \in \ker \delta_\mathcal{G}$. This proves
$$\Phi(\ker \delta_\mathcal{F}) \subseteq \ker \delta_\mathcal{G}.$$

On the other hand, not every global section in $C^0(\mathcal{F})$ lies in the image of global sections in $C^0(\xi)$ under this correspondence. This strict inclusion reflects an "index jump" in the discrete index theory of sheaf-graphs.

**Part 2: Strictness.**

**Lemma A.2** (Strict Inclusion). *If the holonomy of $(G, \mathcal{F})$ is trivial, then*
$$\Phi \ker \mathbf{L}_\mathcal{F} \subsetneq (\ker \mathbf{L}_\mathcal{G}).$$

*Proof.* Under trivial holonomy, any constant section $X_i \equiv P \in \mathrm{SPD}_n$ satisfies $\delta_\mathcal{G}X = I$. For any $X \in \mathrm{Im}(\Phi)$, i.e. $X = xx^T + \varepsilon I_n$ for some $x \in \mathbb{R}^n$, the eigenvalues of $X$ must be $|x|^2 + \varepsilon$ and $\varepsilon$. If $P$ possesses more than two distinct eigenvalues, then $P \notin \mathrm{Im}(\Phi)$, yielding strict inclusion. □

□

## B. Mathematical Interpretations

**Discrete Index Theorem**

**Definition B.1.** The discrete index of a sheaf $(G, \mathcal{F})$ is defined as:
$$\mathrm{Ind}(\mathcal{F}) := \dim \ker \delta_\mathcal{F} - \dim \ker \delta_\mathcal{F}^\top.$$

**Lemma B.2** (Discrete Atiyah–Singer Analogue). *For a connected graph,*
$$\mathrm{Ind}(\mathcal{F}) = (|V| - |E|) \dim \mathcal{F}.$$

*Proof.* The sheaf complex
$$0 \longrightarrow C^0(\mathcal{F}) \xrightarrow{\delta_\mathcal{F}} C^1(\mathcal{F}) \longrightarrow 0$$

is elliptic in the discrete sense. By Proposition 4.8,
$$\mathrm{Ind}(\mathcal{F}) = \dim C^0 - \dim C^1 = (|V| - |E|) \dim \mathcal{F}.$$

□

**Lemma B.3** (Index Jump).
$$\mathrm{Ind}(\mathcal{G}) - \mathrm{Ind}(\mathcal{F}) = (|V| - |E|)\frac{n(n-1)}{2}.$$

This index jump explains the enlargement of $\ker \mathbf{L}_\mathcal{G}$ compared to $\ker \mathbf{L}_\mathcal{F}$.

**Corollary B.4** (Exact Kernel Characterization). *Let $\rho : \pi_1(G) \to O(n)$ denote the holonomy representation, where $\pi_1(G)$ denotes for the fundamental group formed by all cycles on $G$. Then*

$$\ker \mathbf{L}_\mathcal{G} \simeq \big\{ A \in \mathrm{Sym}(n) \mid \rho(\gamma) A \rho(\gamma)^\top = A, \; \forall \gamma \in \pi_1(G) \big\}.$$

*Proof.* A section is harmonic if and only if it is invariant under parallel transport along all cycles, equivalently fixed by the holonomy representation. □

**View of informations under first vs. second order geometry.** The most generally case of dealing with data is viewing it as a *function* at each node, represents the input informations. In mathematical view, one more insightful perspective is to view functions as curves (algebraic curves).

Let $(M, g)$ be a Riemannian manifold with Levi–Civita connection $\nabla$. First-order information is encoded by tangent vectors $X \in T_p M$ and the metric $g$, which determines infinitesimal distances.

For a smooth function $f : M \to \mathbb{R}$, the Hessian $\mathrm{Hess}_f(X, Y) = \langle \nabla_X \nabla f, Y \rangle$ is a symmetric $(0, 2)$-tensor capturing second-order variation; it vanishes iff $f$ is locally affine in normal coordinates. Let $f : M \to \mathbb{R}$ be $C^2$ and $p$ a critical point. The Hessian defines a local bilinear form

$$g_p^{(f)}(X, Y) = \mathrm{Hess}_f(p)(X, Y).$$

If $g_p^{(f)} \succ 0$, then for any curve $\gamma$ with $\gamma(0) = p$, $\dot{\gamma}(0) = v$,

$$f(\gamma(t)) = f(p) + \tfrac{t^2}{2} g_p^{(f)}(v, v) + o(t^2),$$

implying quadratic growth. Thus positive definiteness of the Hessian metric is equivalent to the second-order sufficient condition, and the induced norm $\|v\|_{g^{(f)}}^2 = g_p^{(f)}(v, v)$ measures local curvature of $f$.

The above brief but insightful mathematical perspective prompts us to consider that the prevailing approach to processing node information primarily treats data as vectors. If we interpret this method within the above framework as processing first-order information, then we can equally contemplate an alternative method for processing node information—namely, processing second-order information. This approach requires us to interpret node information as input to a locally defined quadratic form, as previously described, i.e., as a Riemannian metric. This insight leads us to adopt the SPD-Sheaf method.

## C. Experimental Setup

### C.1. Datasets

We evaluate our method on seven molecular property prediction benchmarks from MoleculeNet (Wu et al., 2018), spanning biophysics (BACE, HIV, MUV) and physiology (BBBP, ClinTox, SIDER, Tox21) domains.:

- **BACE**: Binding results for inhibitors of human $\beta$-secretase 1
- **BBBP**: Blood-brain barrier penetration
- **ClinTox**: Clinical trial toxicity
- **SIDER**: Drug side-effects
- **Tox21**: Toxicity measurements
- **HIV**: Inhibition of HIV replication
- **MUV**: Bioactivity data

These datasets span a range of sizes (1,427 to 93,808 molecules) and prediction tasks (1 to 27 binary classification targets), providing a comprehensive evaluation of molecular property prediction capabilities. The original data are SMILES strings. We generate 3D conformers using RDKit with MMFF94 force field optimization, which provides the atomic coordinates required for our coordinate-to-SPD lifting.

### C.2. Baselines

We compare against a comprehensive set of state-of-the-art methods spanning multiple paradigms:

1. **Message-passing neural networks**: D-MPNN (Yang et al., 2019) and AttentiveFP (Xiong et al., 2020)
2. **Pre-training approaches**: N-GramRF/XGB (Liu et al., 2019), PretrainGNN (Hu et al., 2020), GraphMVP (Liu et al., 2022a), MolCLR (Wang et al., 2022b), and Uni-Mol (Zhou et al., 2023)

3. **Geometry-aware models**: GEM (Fang et al., 2022), GROVE (Rong et al., 2020), Mol-GDL (Shen et al., 2023), SchNet (Schütt et al., 2017), EGNN (Satorras et al., 2021)
4. **Transformer-based methods**: SMPT (Li et al., 2024)

## C.3. Evaluation Protocol

Following standard practice, we use scaffold splitting with an 80%/10%/10% train/validation/test partition to ensure molecules with similar scaffolds are grouped together, providing a realistic evaluation of generalization to novel chemical structures. We report the area under the receiver operating characteristic curve (ROC-AUC) averaged over 5 independent runs with different random weight initializations. For multi-task datasets (SIDER, Tox21, MUV), we report the mean ROC-AUC across all tasks. The reported standard deviations reflect variance due to random initialization.

# D. Implementation Details

## D.1. Model Details

**Coordinate-to-SPD Lifting.** Given atomic coordinates $\{p_v\}_{v \in V}$, we first compute the molecular centroid $\bar{p} = \frac{1}{|V|} \sum_v p_v$ and center the coordinates: $u_v = p_v - \bar{p}$. Centering ensures translation invariance of the resulting SPD representation. The centered displacement is normalized to obtain unit direction vectors $\hat{u}_v = \frac{u_v}{\|u_v\| + \epsilon}$, where $\epsilon = 10^{-8}$ ensures numerical stability for atoms located near the molecular centroid (e.g., the central carbon in methane). The initial SPD matrix at each node is then constructed via outer product with regularization:

$$X_v^{(0)} = \hat{u}_v \hat{u}_v^\top + \varepsilon I_3 \in \text{SPD}_3,$$

where $\varepsilon = 10^{-4}$ ensures positive definiteness. This initialization yields effectively rank-1 matrices encoding directional information.

**SPD sheaf convolution layer.** Each geometric layer performs the following operations. Given node SPD features $X_v^{(\ell)} \in \text{SPD}_3$:

*Step 1: Learnable isometry.* Apply a learnable orthogonal transformation:

$$\tilde{X}_v^{(\ell)} = Q^{(\ell)} X_v^{(\ell)} (Q^{(\ell)})^\top,$$

where $Q^{(\ell)} \in \text{O}(3)$ is parameterized via QR decomposition of learnable parameters $W^{(\ell)} \in \mathbb{R}^{3 \times 3}$:

$$W = QR, \quad Q^{(\ell)} = Q \cdot \text{sign}(\text{diag}(R)).$$

*Step 2: Sheaf Laplacian computation.* For each edge $e = (u, v)$, compute orthogonal restriction maps $M_{ue}^{(\ell)}, M_{ve}^{(\ell)} \in \text{O}(3)$ via the sheaf learner. The coboundary in the log domain is:

$$T_e = \log(M_{ve}^{(\ell)} \tilde{X}_v^{(\ell)} M_{ve}^{(\ell)\top}) - \log(M_{ue}^{(\ell)} \tilde{X}_u^{(\ell)} M_{ue}^{(\ell)\top}).$$

The adjoint aggregates back to nodes with pullback transformations:

$$\Delta_v = \sum_{e:v \in e} \pm M_{ve}^{(\ell)\top} T_e M_{ve}^{(\ell)},$$

where the sign depends on edge orientation. Eigenvalues of $\Delta_v$ are normalized to $[-1, 1]$ for stability.

*Step 3: Residual update.* The SPD feature is updated via:

$$X_v^{(\ell+1)} = \text{TgReEig}\left(\exp\left(\log X_v^{(\ell)} + \Delta_v\right)\right).$$

**Restriction map parameterization.** Orthogonal restriction maps are learned from semantic node features via a quadratic form sheaf learner (Bodnar et al., 2022). For edge $e = (u, v)$:

$$s_{ue}^{(\ell)} = \text{MLP}_{\text{sheaf}}([h_u^{(\ell)} \| h_v^{(\ell)}]) \in \mathbb{R}^6,$$

where $s_{ue}^{(\ell)}$ is reshaped into the lower triangular part of a $3 \times 3$ matrix $L_{ue}^{(\ell)}$. The skew-symmetric matrix is then constructed as $S_{ue}^{(\ell)} = L_{ue}^{(\ell)} - L_{ue}^{(\ell)\top} \in \mathbb{R}^{3 \times 3}$. The orthogonal matrix is obtained via the Cayley transform:

$$M_{ue}^{(\ell)} = (I - S_{ue}^{(\ell)}/2)^{-1}(I + S_{ue}^{(\ell)}/2) \in \text{O}(3),$$

which guarantees exact orthogonality throughout training.

**SPD nonlinearity (TgReEig).** We apply nonlinearity in the eigenvalue domain. Given $X = V\Lambda V^\top$ with $\Lambda = \mathrm{diag}(\lambda_1, \ldots, \lambda_d)$:

$$\mathrm{TgReEig}(X) = V \cdot \mathrm{diag}(\tilde{\lambda}_1, \ldots, \tilde{\lambda}_d) \cdot V^\top,$$

where

$$\tilde{\lambda}_i = \begin{cases} \exp(\log \lambda_i) & \text{if } \log \lambda_i > 0 \\ \exp(\delta \cdot i) & \text{otherwise} \end{cases}$$

with $\delta = 0.1$ providing step noise to maintain positive definiteness.

**Achieving Rotation Invariance via Equivariant Local Frames.** The initial SPD representation $X_v^{(0)} = \hat{u}_v \hat{u}_v^\top + \varepsilon I_3$ is O(3)-equivariant: under a global rotation $R \in \mathrm{O}(3)$, it transforms as $X_v^{(0)} \mapsto R X_v^{(0)} R^\top$. For molecular property prediction tasks considered in this work, the target properties (e.g., toxicity, solubility) are intrinsic to the molecule and should be invariant to rigid transformations. We therefore canonicalize this equivariant representation to achieve invariance. We note that the natural equivariance of our SPD embedding could be directly leveraged for equivariant prediction tasks such as molecular force or stress tensor prediction, which we leave for future work.

To achieve rotation invariance, we construct a local orthonormal frame $M_v \in \mathrm{O}(3)$ for each node that is also equivariant under global rotations. Specifically, we build $M_v = [v_1, v_2, v_3]$ using:

1. $v_1 = u_v / \|u_v\|$: the normalized displacement from the molecular centroid;
2. $v_2$: the orthogonalized aggregated neighbor edge directions, computed via Gram-Schmidt;
3. $v_3 = v_1 \times v_2$: the cross product completing the orthonormal basis.

Under a global rotation $R$, this frame transforms equivariantly as $M_v \mapsto R M_v$.

At the first layer, we apply the local frame transformation to obtain rotation-invariant input:

$$\bar{X}_v^{(0)} = M_v^\top X_v^{(0)} M_v.$$

This representation is rotation invariant since:

$$M_v^\top X_v^{(0)} M_v \ \mapsto \ (R M_v)^\top (R X_v^{(0)} R^\top)(R M_v) = M_v^\top X_v^{(0)} M_v.$$

**Proposition D.1** (Invariance Preservation). *If the input to a SPD sheaf convolution layer is rotation invariant, then the output is also rotation invariant.*

*Proof.* Let $X_v^{(\ell)}$ be rotation invariant for all $v \in V$. The SPD sheaf layer consists of the following operations:

1. **Learnable isometry:** $\tilde{X}_v^{(\ell)} = Q^{(\ell)} X_v^{(\ell)} Q^{(\ell)\top}$, where $Q^{(\ell)} \in \mathrm{O}(3)$ is a learned parameter independent of the input coordinate system. Since $X_v^{(\ell)}$ is invariant and $Q^{(\ell)}$ does not depend on 3D coordinates, $\tilde{X}_v^{(\ell)}$ remains invariant.
2. **Restriction maps:** $\mathcal{F}_{v \to e}(\tilde{X}_v^{(\ell)}) = M_{ve}^{(\ell)} \tilde{X}_v^{(\ell)} M_{ve}^{(\ell)\top}$, where $M_{ve}^{(\ell)} \in \mathrm{O}(3)$ is predicted from semantic node features $h_v^{(\ell)}, h_u^{(\ell)}$ via the sheaf learner (Equation 23). Since semantic features encode discrete chemical information (atom types, bond types) that is inherently rotation invariant, the restriction maps $M_{ve}^{(\ell)}$ are invariant. Combined with invariant $\tilde{X}_v^{(\ell)}$, the edge representations remain invariant.
3. **Coboundary and adjoint:** These operations (Equations 5, 7) involve matrix logarithms, additions in $\mathrm{Sym}_3$, and exponentials—all coordinate-free algebraic operations that preserve invariance.
4. **Lie group update:** $X_v^{(\ell+1)} = X_v^{(\ell)} \odot (\mathbf{L}_{\mathcal{F}}^{(\ell)} \tilde{X}^{(\ell)})_v$ (Equation 15) combines invariant quantities via the Lie group operation, yielding an invariant output.

By induction, all layers produce rotation-invariant representations. $\square$

Since $\bar{X}_v^{(0)}$ is rotation invariant after the first-layer canonicalization, all subsequent SPD representations $X_v^{(\ell)}$ and the final pooled representation $\bar{X}_G$ are rotation invariant, ensuring that the model's predictions are independent of the arbitrary 3D orientation of the input molecule.

**Cross-modal interaction.** Let $X_v^{(\ell)} \in \mathrm{SPD}_3$ denote the geometric feature and $h_v^{(\ell)} \in \mathbb{R}^{d_f}$ the semantic feature. The interaction proceeds as:

*Step 1:* Project SPD to tangent space and vectorize:

$$\xi_v = \mathrm{vec}(\log X_v^{(\ell)}) \in \mathbb{R}^9.$$

*Step 2:* Compute attention-weighted modulation:

$$\alpha_v = \sigma \left( \mathrm{MLP}_{\mathrm{attn}} \left( [W_{\mathrm{spd}}\xi_v \| W_{\mathrm{feat}} h_v^{(\ell)}] \right) \right),$$

$$h_v^{(\ell)} \leftarrow h_v^{(\ell)} + \alpha_v \cdot W_\xi \xi_v,$$

where $W_\xi \in \mathbb{R}^{d_f \times 9}$ projects geometric information to the semantic dimension. This design allows geometry to modulate semantic features while keeping the SPD stream intrinsic.

**Semantic stream.** The semantic stream processes chemical features via GraphSAGE with mean aggregation:

$$h_v^{(\ell+1)} = \sigma \left( W^{(\ell)} \cdot \mathrm{CONCAT} \left( h_v^{(\ell)}, \mathrm{MEAN}_{u \in \mathcal{N}(v)} h_u^{(\ell)} \right) \right),$$

where $\sigma$ denotes LeakyReLU activation. Input node features are encoded using CGCNN encoding (Xie & Grossman, 2018) with dimension 92.

**SPD pooling.** For graph-level prediction, we aggregate node SPD matrices via power-Euclidean pooling (Chen et al., 2025):

$$\bar{X}_G = \left( \frac{1}{|V|} \sum_{v \in V} (X_v^{(L)})^\theta \right)^{1/\theta},$$

where $\theta = 0.5$ (matrix square root). Matrix powers are computed via eigendecomposition.

**Fusion and prediction head.** The pooled SPD matrix is mapped to a geometric descriptor: $g = \mathrm{vec}(\log \bar{X}_G)$ The semantic stream produces $h_G \in \mathbb{R}^{d_f}$ via mean pooling. For most datasets, we use Factorized Bilinear Pooling (FBP):

$$z = (Ug) \odot (Vh_G), \quad \hat{y} = \mathrm{MLP}(\mathrm{BN}([g\|h_G\|z])).$$

For BACE, we use Cross-Attention fusion where semantic features query geometric features via multi-head attention ($d_{\mathrm{model}} = 64$, 4 heads), followed by residual connections and layer normalization.

On BACE, all methods in Table 2 (including Euclidean SheafNN and other baselines) use Cross-Attention fusion to ensure a fair comparison. Under this controlled setting, SPD-Sheaf (89.0%) outperforms Euclidean SheafNN (87.5%) by 1.5%, demonstrating the contribution of SPD geometry independent of the fusion head.

We additionally evaluated SPD-Sheaf with FBP fusion, which achieves 87.3%. This indicates that Cross-Attention provides a further 1.7% improvement on BACE, likely due to the importance of fine-grained geometric-semantic interaction in protein-ligand binding prediction. For all other datasets, FBP provides sufficient capacity.

**Numerical stability.** Matrix logarithms and exponentials are computed via eigendecomposition. Eigenvalues are clamped to $[\varepsilon, +\infty)$ with $\varepsilon = 10^{-4}$. All SPD-valued computations use `float64` precision.

### D.2. Training

**Optimizer.** We use a hybrid optimization strategy based on parameter tensor dimensionality:

- **Muon optimizer** (Jordan et al., 2024) for parameters with $\mathrm{ndim} \geq 2$ (weight matrices); learning rate $\eta_{\mathrm{Muon}} = 2 \times 10^{-2}$.
- **Adam optimizer** for parameters with $\mathrm{ndim} < 2$ (biases, normalization); learning rate $\eta_{\mathrm{Adam}} = 5 \times 10^{-4}$, $\beta = (0.9, 0.95)$.

Weight decay is set to $10^{-2}$ for all parameters. No learning rate scheduler is used.

**Loss function.** Binary cross-entropy loss:

$$\mathcal{L} = -\frac{1}{N} \sum_{i=1}^{N} [y_i \log \hat{y}_i + (1 - y_i) \log(1 - \hat{y}_i)].$$

For multi-task datasets (SIDER, Tox21, MUV), the loss is averaged across all tasks.

**Regularization.** Dropout rate 0.1 applied to semantic stream and SPD tangent space features. Gradient clipping with maximum norm 3.0.

**Dataset-specific hyperparameters.** Table 5 summarizes the key hyperparameters.

*Table 5.* Dataset-specific hyperparameters.

| Dataset | Layers | Batch Size | Fusion Type | Epochs |
|---------|--------|------------|-------------|--------|
| BACE | 2 | 64 | Cross-Attention | 200 |
| BBBP | 2 | 128 | FBP | 200 |
| ClinTox | 5 | 128 | FBP | 200 |
| SIDER | 2 | 128 | FBP | 200 |
| Tox21 | 7 | 512 | FBP | 200 |
| HIV | 5 | 512 | FBP | 200 |
| MUV | 7 | 512 | FBP | 200 |

Common hyperparameters: semantic feature dimension $d_f = 64$, SPD matrix dimension $d = 3$, FBP rank $r = 64$, dropout 0.1, $\varepsilon = 10^{-4}$, power-Euclidean pooling $\theta = 0.5$.

**Data splitting.** We use scaffold splitting with 80%/10%/10% train/validation/test partition, ensuring molecules with similar scaffolds are grouped together.

### D.3. Hardware and Runtime

All experiments were conducted on a single NVIDIA A100-SXM4-40GB GPU.

### D.4. Software Dependencies

Our implementation uses: PyTorch 2.0+ (`float64` default dtype), DGL for graph operations, RDKit for molecular processing and 3D conformer generation, pytorch-optimizer for Muon, and scikit-learn for evaluation metrics.

Code is available at `https://github.com/YuhanPeng0524/SPDSheaf`.

## E. EEG-Based Motor Imagery Experiment

In Electroencephalography (EEG)-based motor imagery classification, EEG signals are collected while users mentally rehearse specific movements (e.g., left/right hand, foot, or tongue) without any overt execution (Pfurtscheller & Neuper, 2001). For a comprehensive overview of SPD matrix learning methods in neuroimaging, we refer to Ju et al. (2025). Such imagery modulates sensorimotor rhythms, most prominently in the mu and beta bands, and produces characteristic event-related desynchronization/synchronization (ERD/ERS) patterns over sensorimotor regions. These ERD/ERS responses serve as class-discriminative neural signatures that can be translated into control commands for BCIs.

In this work, we adopt the *time-frequency graph* framework (Ju & Guan, 2024) for EEG classification. Unlike conventional EEG graph construction where scalp electrodes are treated as graph vertices, each node in the time-frequency graph is defined as an EEG spatial covariance matrix computed within a specific time interval and frequency band, yielding SPD matrix-valued node features. This formulation naturally casts the learning problem as graph classification, which aligns with the overall framework proposed in this work.

### E.1. Construction of Time-Frequency Graph

Given a time-frequency segmentation plan (with optional overlap) $\{\Delta t_i \times \Delta f_i\}_{i \in \mathcal{I}}$, the EEG is first band-pass filtered within each frequency interval $\Delta f_i$, and the corresponding temporal segment $X_i \in \mathbb{R}^{n_C \times \Delta t_i}$ is extracted. We use Chebyshev Type II filters with a maximum passband loss of 3 dB and minimum stopband attenuation of 30 dB. An SPD matrix for each segment is then formed via the sample covariance $S(\Delta t_i \times \Delta f_i) := X_i X_i^\top$, and $\{S(\Delta t_i \times \Delta f_i)\}_{i \in \mathcal{I}}$ is collected as a time-frequency distribution on the SPD manifold.

The segmentation is selected to align with event-related modulations of sensorimotor rhythms (e.g., ERD/ERS), which are known to be frequency-specific and user-dependent; therefore, the time and frequency discretizations may be varied across experimental settings. In practice, $\Delta f$ is often chosen with reference to canonical neurophysiological bands (e.g., $\delta$, $\theta$, $\mu$, $\beta$, $\gamma$), while flexible refinement is allowed to better capture subject-specific discriminative responses.

To characterize the local structure in this SPD matrix-valued time-frequency distribution, a time-frequency graph $\mathcal{G}(\mathcal{V}, \mathcal{E})$ is constructed, where each vertex is associated with one covariance matrix $S_i$. Edges are determined by a locality-constrained $\epsilon$-neighborhood rule: two vertices $S_i = S(\Delta t_i \times \Delta f_i)$ and $S_j = S(\Delta t_j \times \Delta f_j)$ are connected if they are located within a local time-frequency window $\mathcal{B}_{\epsilon_1, \epsilon_2}$ and are sufficiently close under the AIRM Riemannian metric. The window constraint is imposed to enforce locality along the time-frequency plane with width $\epsilon_1 \geq 0$ in time and height $\epsilon_2 \geq 0$ in frequency:

$$0 \leq \mathrm{mid}(\Delta t_i) - \mathrm{mid}(\Delta t_j) \leq \epsilon_1,$$
$$\left| \mathrm{mid}(\Delta f_i) - \mathrm{mid}(\Delta f_j) \right| \leq \epsilon_2,$$

where $\mathrm{mid}(\cdot)$ denotes the midpoint of an interval; the one-sided temporal constraint is adopted to reflect the forward direction of time evolution. Among pairs satisfying the window constraint, edges are retained for those satisfying $d_{g^{\mathrm{AIRM}}}(S_i, S_j)^2 < \epsilon$. Finally, the adjacency matrix $A$ is defined using a radial basis function (RBF) kernel weighting on the AIRM Riemannian distance between adjacent vertices:

$$A_{ij} = \begin{cases} \exp\left(-d^2_{g^{\mathrm{AIRM}}}(S_i, S_j)/t\right), & \text{if } (S_i, S_j) \in \mathcal{E}, \\ 0, & \text{otherwise}, \end{cases}$$

where $t > 0$ is a preset kernel bandwidth.

### E.2. Experimental Dataset

We evaluate our approach on the BNCI2015_001 motor-imagery EEG dataset from the BNCI Horizon 2020 initiative, accessed through the MOABB[1] repository. The dataset contains recordings from 12 subjects performing two sustained motor imagery tasks: right-hand imagery and both-feet imagery. EEG was collected with a g.GAMMAsys active electrode system and a g.USBamp amplifier (g.tec, Graz, Austria) at 512 Hz, with online preprocessing consisting of a 0.5-100 Hz band-pass filter and a 50 Hz notch filter.

**Channel montage.** The recordings are provided as Laplacian derivations centered at C3, Cz, and C4 (International 10-20 system), using a 3.5 cm center-to-center layout around each center site. The Laplacian derivations are formed from the surrounding electrodes: C3 (FC3, C5, CP3, C1), Cz (FCz, C1, CPz, C2), and C4 (FC4, C2, CP4, C6). Channels are ordered as FC3, FCz, FC4, C5, C3, C1, Cz, C2, C4, C6, CP3, CPz, CP4.

**Trial paradigm and time window.** Each trial starts with a 3 s reference period, followed by an auditory cue and a visual cue presented from 3.0 to 4.25 s (right arrow for right-hand imagery; down arrow for both-feet imagery). A feedback-driven activity period then spans 4.0-8.0 s, and consecutive trials are separated by a random 2-3 s inter-trial interval. In accordance with the protocol, subjects perform motor imagery from cue onset (3.0 s) until the end of the trial (8.0 s); we therefore use the 3.0-8.0 s segment (5 s) as the primary analysis window.

**Sessions and number of trials.** Subjects 1-8 were recorded across two sessions (A, B) on consecutive days, while subjects 9-12 included an additional third session (C). Each session contains 100 trials per class (200 trials/session). Overall, the dataset comprises 28 sessions and 5,600 trials ($200 \times 28$), providing sufficient data for representation learning and downstream evaluation. In our experiments, we restrict all subjects to the first two sessions (A and B) only.

### E.3. Experimental Scenarios

We consider two widely used evaluation protocols for EEG-based motor imagery classification:

1. **10-fold Cross-Validation (CV).** For each subject, trials are partitioned into 10 equal-sized, class-balanced folds. We train on 9 folds and evaluate on the remaining fold, repeating the procedure 10 times such that each fold serves as the test set once.
2. **Cross-Session Holdout.** To assess robustness to inter-session (between-days) variability, we adopt a session-wise split: models are trained on the first session (A) and evaluated on the entire second session (B). As sessions A and B are typically recorded on different days, this protocol captures session-to-session distribution shifts.

Note that, under both the 10-fold CV and holdout scenarios, a graph is constructed and its edge weights are computed using only the training portion of each EEG trial. In particular, no data from the test split is used to construct the graph.

---

[1] https://moabb.neurotechx.com/docs/generated/moabb.datasets.BNCI2015_001.html#moabb.datasets.BNCI2015_001

*Table 6.* Classification accuracy (%) on the BNCI2015_001 motor imagery dataset across three evaluation protocols. Mean (std) reported where standard deviations are available. **Bold** indicates best; underline indicates second-best.

| Scenario | FBCSP | FBCNet | Tensor-CSPNet | TSMNet | GyroAtt | Graph-CSPNet | SPD-Sheaf (Ours) |
|---|---|---|---|---|---|---|---|
| 10-Fold CV (A) | 79.46 (14.16) | 82.62 (13.11) | 81.29 (14.78) | 82.17 | 82.33 | **84.62** | 83.25 (11.29) |
| 10-Fold CV (B) | 81.96 (11.14) | 84.92 (10.30) | 85.29 (10.54) | 84.67 | 86.04 | **88.00** | 85.58 (9.76) |
| Holdout (A → B) | 73.46 (14.09) | 74.50 (16.01) | 79.04 (14.67) | 77.08 | 78.46 | 79.75 | **82.58** (12.15) |

## E.4. SPD-Sheaf Configuration for EEG Classification

For EEG-based motor imagery classification, we apply the proposed SPD sheaf framework on each time-frequency graph without a semantic stream, as the node features are inherently SPD matrix-valued.

**SPD-Sheaf Architecture.** Each node in the time-frequency graph carries an SPD matrix $X_v \in \mathrm{SPD}_{13}$ (e.g., spatial covariance matrix derived from 13-channel motor imagery EEGs). Hence, we apply the proposed SPD sheaf convolution, which is defined in Equation 15, as follows:

$$ X_v^{(\ell+1)} = \mathrm{TgReEig}\left( X_v^{(\ell)} \odot \exp\left( \sum_{(v,u)=e} (-1)^{I(v,e)} M_{ve}^{(\ell)\top} \left( \log(\mathcal{F}_{v\to e}^{(\ell)} X_v^{(\ell)}) - \log(\mathcal{F}_{u\to e}^{(\ell)} X_u^{(\ell)}) \right) M_{ve}^{(\ell)} \right) \right), $$

where the restriction maps $\mathcal{F}_{v\to e}(P) = M_{ve} P M_{ve}^\top$ with learnable $M_{ve} \in O(13)$.

For the orthogonal restriction map, we derive Euclidean node features from the SPD matrices via the log-Euclidean vectorization:

$$ \mathbf{h}_v = \mathrm{vec}_{\mathrm{upper}}(\log X_v) \in \mathbb{R}^{91}, $$

where $91 = 13 \times 14/2$ is the number of upper-triangular elements.

For each edge $e = (u, v)$, the restriction map parameters are predicted from the concatenated features $[\mathbf{h}_u \| \mathbf{h}_v]$ via an MLP, then converted to orthogonal matrices $M_{ue}, M_{ve} \in O(13)$ through the Cayley transform.

The model adopts two SPD sheaf layers, trained for 50 epochs with a learning rate of $3 \times 10^{-4}$ and a batch size of 16.

## E.5. Results

Table 6 reports classification accuracy on BNCI2015_001 against six baselines spanning classical signal-processing methods (FBCSP (Ang et al., 2008), FBCNet (Mane et al., 2021)) and recent SPD-based deep learning methods (Tensor-CSPNet (Ju & Guan, 2023), TSMNet (Kobler et al., 2022), GyroAtt (Wang et al., 2025), Graph-CSPNet (Ju & Guan, 2024)). Graph-CSPNet results are quoted from Ju & Guan (2024); GyroAtt and TSMNet are re-run under the identical protocol.

We emphasize that SPD-Sheaf was designed for molecular property prediction and is applied here *without any task-specific modification*: it operates directly on the time-frequency graph of Ju & Guan (2024), with no semantic stream and no coordinate-to-SPD lifting. Despite this, SPD-Sheaf attains the best cross-session generalization score (Holdout A→B: 82.58% vs. 79.75% for Graph-CSPNet, +2.83 points). On the within-session 10-fold CV protocol, SPD-Sheaf is competitive (second on session A, third on session B), surpassed by Graph-CSPNet, which is specifically engineered for the time-frequency EEG setting.

The strong cross-session result suggests that sheaf-theoretic edge heterogeneity provides robust generalization under distribution shift, even without domain-specific tuning. We therefore position the EEG experiments as supplementary validation of the framework's generality rather than its primary application; a fully EEG-tailored variant is left to future work.

**Discussion (Key Difference from Molecular Experiments).** Unlike the molecular setting where SPD matrices are lifted from 3D coordinates (effectively rank-1 initialization in $\mathrm{SPD}_3$), EEG spatial covariance matrices are derived from multi-channel EEGs, yielding higher-rank initial representations in $\mathrm{SPD}_{13}$. This demonstrates the versatility of our framework: the same SPD sheaf convolution operates natively on domains where second-order statistics arise naturally, without requiring the coordinate-to-SPD lifting or dual-stream architecture.

# F. Regression Experiment

## F.1. ZINC Molecular Property Regression

While the main experiments focus on binary classification tasks from MoleculeNet, we additionally evaluate our SPD-Sheaf framework on graph-level regression to demonstrate its versatility across prediction paradigms. We use the ZINC benchmark, a standard dataset for evaluating graph neural networks on molecular property prediction.

**Dataset Description.** **ZINC** is a subset of the ZINC database containing 12,000 drug-like molecules with the official split from the Benchmarking GNNs paper: 10,000 training, 1,000 validation, and 1,000 test molecules (Irwin & Shoichet, 2005; Sterling & Irwin, 2015; Dwivedi et al., 2023). The regression target is the **penalized logP** (constrained solubility), defined as:

$$\text{penalized logP} = \log P - \text{SAS} - \text{ring\_penalty}$$

where $\log P$ is the octanol-water partition coefficient (Wildman-Crippen method), SAS is the synthetic accessibility score, and $\text{ring\_penalty} = \sum_{r:|r|>6}(|r| - 6)$ penalizes rings with more than 6 atoms.

This property is geometry-sensitive: molecules with identical atom types and connectivity can have different penalized logP values due to conformational differences affecting solubility and synthetic accessibility.

**Experimental Setup.** **Data preparation.** Since SPD-Sheaf GNN requires 3D molecular structures to compute geometric shortest path distances, we recover SMILES strings for the ZINC 12K split by matching molecules from TDC ZINC250K using molecular formula fingerprints, achieving 92.5% coverage (11,094/12,000 molecules). 3D conformers are generated using RDKit with MMFF force field optimization. To ensure fair comparison, all baseline models in Table 7 are retrained on this identical filtered dataset with same hyperparameters used in (Dwivedi et al., 2023). Note that SPD-Sheaf uses RDKit-derived atom and bond features (92 and 21 dimensions respectively) to leverage the recovered SMILES information, while baseline models use the original ZINC atom/bond type embeddings.

**Architecture.** We use the same dual-stream SPD-Sheaf architecture as in classification experiments except for the output head.

**Regression head.** The fused representation $z = [g\|h_G\|(Ug) \odot (V h_G)]$ is passed through an MLP regressor outputting a scalar prediction. We use L1 loss (Mean Absolute Error) for training.

**Hyperparameters.**

- Layers: 5 (geometric) / 5 (semantic)
- Hidden dimension: 128
- SPD matrix size: $3 \times 3$
- Batch size: 256
- Weight decay: 0.01
- Iterations: 4
- Epochs: 200
- Optimzer: Same as main experiment

**Evaluation metric.** Mean Absolute Error (MAE) on test set, lower is better. We report mean and standard deviation over 4 runs for SPD-Sheaf model.

## F.2. Baseline Methods

**Baseline selection.** We compare against representative message-passing neural networks from the Benchmarking GNNs framework (Dwivedi et al., 2023). We exclude higher-order Weisfeiler-Lehman methods (3WLGNN and RingGNN) from our comparison as their $\mathcal{O}(n^3)$ dense tensor operations result in prohibitively slow training (~25 minutes per epoch), preventing convergence within the standard 12-hour computational budget used in the original benchmark. This limitation was also noted by Dwivedi et al. (2023), who reported training difficulties and divergence issues with these architectures.

## F.3. Analysis

**SPD-Sheaf achieves competitive performance.** As shown in Table 8, SPD-Sheaf (MAE 0.258) outperforms most standard message-passing methods despite using only 5 layers compared to 16 layers for baselines. Specifically, it improves over GCN (0.274), MoNet (0.279), GatedGCN (0.313), GraphSage (0.315), GAT (0.324), and GIN (0.422). SPD-Sheaf achieves comparable performance to GatedGCN-E (0.254), which explicitly incorporates edge features. This demonstrates that our geometric SPD representations effectively capture molecular structure information relevant for property prediction.

*Table 7.* Baseline methods for ZINC regression.

| Method | Type | Reference |
|---|---|---|
| GCN | Message-passing | Kipf & Welling (2017) |
| GraphSage | Sampling-based | Hamilton et al. (2017) |
| GIN | Message-passing | Xu et al. (2019) |
| GAT | Attention-based | Veličković et al. (2018) |
| MoNet | Gaussian mixture | Monti et al. (2017) |
| GatedGCN | Gated edges | Bresson & Laurent (2017) |
| GatedGCN-E | + Edge features | Bresson & Laurent (2017) |
| GatedGCN-E-PE | + Positional encoding | Dwivedi et al. (2023) |

*Table 8.* ZINC regression results on filtered dataset. Test MAE $\pm$ s.d. (lower is better), averaged over 4 runs. The best performance is marked in **bold**, and the second best is underlined.

| Model | $L$ | Test MAE ↓ |
|---|---|---|
| GCN | 16 | $0.274 \pm 0.006$ |
| MoNet | 16 | $0.279 \pm 0.015$ |
| GatedGCN | 16 | $0.313 \pm 0.007$ |
| GraphSage | 16 | $0.315 \pm 0.005$ |
| GAT | 16 | $0.324 \pm 0.007$ |
| GIN | 16 | $0.422 \pm 0.037$ |
| GatedGCN-E | 16 | $\underline{0.254 \pm 0.004}$ |
| GatedGCN-E-PE | 16 | $\mathbf{0.212 \pm 0.009}$ |
| **SPD-Sheaf (Ours)** | 5 | $0.258 \pm 0.007$ |

**Comparison with positional encoding methods.** GatedGCN-E-PE (0.212) achieves the best performance by combining edge gating with Laplacian positional encodings. The gap between SPD-Sheaf and GatedGCN-E-PE suggests that learned positional encodings capture complementary structural information beyond what 3D geometric structure provide. We note that our method uses *explicit* 3D coordinates from conformer generation, while positional encodings are *learned* spectral features from the graph Laplacian. Future work could explore combining SPD-Sheaf with learned positional encodings to leverage both geometric and spectral information.

## G. Scaling to Larger Molecular Datasets

The MoleculeNet benchmarks evaluated in Section 6 range from 1.4k to 93k molecules. To assess whether SPD-Sheaf scales to substantially larger datasets, we additionally evaluate on **ogbg-molpcba** from the Open Graph Benchmark, which contains 437k molecules—approximately $4.7\times$ the size of MUV and $10.6\times$ the size of HIV.

**Setup.** We apply SPD-Sheaf without any task-specific architectural modification. Recall that the model is a dual-stream architecture combining a geometric stream (SPD sheaf convolution over 3D coordinates) with a semantic stream (GCN over chemical features). For a controlled comparison we report a GCN-only baseline alongside the full dual-stream SPD-Sheaf, both trained under identical conditions. We follow the standard OGB scaffold split and report Average Precision (AP), the official metric for ogbg-molpcba.

**Results.** Table 9 reports the results. SPD-Sheaf attains $AP = 0.2419$, a $19.8\%$ relative improvement over the GCN baseline. This indicates that the gains from second-order geometric representations and edge-specific restriction maps persist at scale, without task-specific tuning.

## H. Additional Oversmoothing Analysis

To complement the depth-robustness analysis in Section 6.5, we quantitatively measure representation collapse using two standard oversmoothing diagnostics.

*Table 9.* Performance on OGBG-MOLPCBA. We report Average Precision (higher is better) on the official scaffold split.

| Model | Avg. Precision |
|---|---|
| GCN (baseline) | 0.2020 |
| SPD-Sheaf (Ours) | **0.2419** (+19.8%) |

*Table 10.* Oversmoothing metrics across depths on BACE (molecular), Cora (homophilic), and Texas (heterophilic). GCN collapses to near zero; SPD-Sheaf maintains $\mathcal{E}_{\mathrm{Dir}}/N > 1$ across all depths and datasets.

| Dataset | Model | Metric | Depth | | | | | |
|---|---|---|---|---|---|---|---|---|
| | | | 2 | 8 | 16 | 32 | 64 | 128 |
| BACE | GCN | $\mathcal{E}_{\mathrm{Dir}}/N$ | 0.594 | 0.126 | 0.108 | 0.016 | 0.002 | 0.000 |
| | | MAD | 0.294 | 0.052 | 0.041 | 0.006 | 0.001 | 0.000 |
| | SPD-Sheaf | $\mathcal{E}_{\mathrm{Dir}}/N$ | 1.280 | 1.719 | 1.687 | 1.715 | 1.515 | 1.125 |
| | | MAD | 0.583 | 0.768 | 0.780 | 0.786 | 0.715 | 0.479 |
| Cora | GCN | $\mathcal{E}_{\mathrm{Dir}}/N$ | 0.028 | 0.006 | 0.000 | 0.000 | 0.000 | 0.001 |
| | | MAD | 0.007 | 0.001 | 0.000 | 0.000 | 0.000 | 0.000 |
| | SPD-Sheaf | $\mathcal{E}_{\mathrm{Dir}}/N$ | 2.530 | 3.014 | 2.980 | 2.945 | 2.543 | 2.064 |
| | | MAD | 0.531 | 0.658 | 0.664 | 0.678 | 0.588 | 0.452 |
| Texas | GCN | $\mathcal{E}_{\mathrm{Dir}}/N$ | 1.371 | 0.114 | 0.010 | 0.001 | 0.000 | 0.000 |
| | | MAD | 0.435 | 0.037 | 0.003 | 0.000 | 0.000 | 0.000 |
| | SPD-Sheaf | $\mathcal{E}_{\mathrm{Dir}}/N$ | 1.915 | 2.337 | 2.571 | 2.316 | 2.843 | 1.797 |
| | | MAD | 0.544 | 0.684 | 0.733 | 0.703 | 0.678 | 0.567 |

**Metrics.** We follow standard practice for quantifying oversmoothing in graph neural networks. The normalized Dirichlet energy $\mathcal{E}_{\mathrm{Dir}}/N$ measures pairwise feature dissimilarity between connected nodes, while the Mean Average Distance (MAD) measures the average cosine distance between connected node pairs (Rusch et al., 2023; Zhang et al., 2026). Both metrics approach 0 as neighboring representations homogenize.

**Setup.** We sweep depths $\{2, 8, 16, 32, 64, 128\}$ on three datasets covering distinct regimes: BACE (molecular graphs, MoleculeNet), Cora (homophilic node classification), and Texas (heterophilic node classification). All other hyperparameters follow the main experiments.

**Results.** Table 10 reports the full numerical values. GCN collapses exponentially: $\mathcal{E}_{\mathrm{Dir}}/N$ drops below $10^{-3}$ by depth 64 on BACE and Texas, and is already below $10^{-2}$ at depth 2 on Cora. In contrast, SPD-Sheaf maintains $\mathcal{E}_{\mathrm{Dir}}/N > 1$ at every depth on every dataset, with MAD remaining above $0.45$ even at depth 128. This empirically confirms our theoretical claim: edge-specific orthogonal restriction maps structurally enlarge $\ker \mathbf{L}_{\mathcal{G}}$, preventing the convergence to constant cochains that drives oversmoothing in standard message passing.

## I. Graph Perturbation Analysis

A natural question is whether SPD-Sheaf's empirical gains genuinely originate from modeling edge-dependent structure, as opposed to other architectural factors such as increased capacity or the dual-stream design. We address this with a controlled graph-perturbation experiment whose purpose is *mechanistic verification* rather than benchmark robustness.

If a model truly leverages edge-specific structure, then corrupting edges should disproportionately degrade its performance. Conversely, a model that relies primarily on node features (and uses edges only as a generic aggregation scaffold) will be relatively insensitive to which edges are present. We therefore measure retention after edge corruption:

$$\mathrm{Retention}(p) := \frac{\mathrm{ROC\text{-}AUC}(p)}{\mathrm{ROC\text{-}AUC}(p=0)} \times 100\%,$$

where $p$ is the edge perturbation rate. *Lower* retention under matched architecture indicates *stronger* reliance on edge-dependent structure.

**Setup.** We consider two perturbation types:

*Table 11.* Graph perturbation analysis. Values are retention (% of clean ROC-AUC) at perturbation rate $p$. SPD-Sheaf shows consistently lower retention than Euclidean SheafNN under matched architecture across both datasets and both perturbation types, indicating stronger reliance on edge-dependent structure.

| Dataset | Perturbation | $p = 0.1$ | | | $p = 0.3$ | | | $p = 0.5$ | | |
|---------|-------------|-----------|-----------|------|-----------|-----------|------|-----------|-----------|------|
| | | SPD-Sheaf | Euc. Sheaf | GCN | SPD-Sheaf | Euc. Sheaf | GCN | SPD-Sheaf | Euc. Sheaf | GCN |
| BACE | Drop | 96.3 | 98.4 | 92.4 | 82.5 | 92.3 | 85.3 | 69.5 | 83.0 | 73.3 |
| | Rewire | 90.1 | 94.7 | 88.5 | 80.1 | 85.8 | 79.0 | 69.5 | 76.4 | 76.0 |
| BBBP | Drop | 98.0 | 99.3 | 98.9 | 92.3 | 98.0 | 96.7 | 89.2 | 94.1 | 97.1 |
| | Rewire | 94.0 | 97.1 | 101.6 | 86.6 | 97.4 | 98.8 | 83.8 | 91.5 | 98.4 |

- **Drop:** each edge is independently removed with probability $p$.
- **Rewire:** each edge is independently rewired to a random endpoint with probability $p$, preserving the degree distribution in expectation.

We evaluate $p \in \{0.1, 0.3, 0.5\}$ on BACE and BBBP, comparing three architectures under the same dual-stream framework: SPD-Sheaf (ours); Euclidean SheafNN, which replaces SPD stalks with vector stalks while keeping all other components identical; and GCN, which removes both sheaf structure and SPD geometry. The Euclidean SheafNN comparison is the key control: it isolates the contribution of SPD-valued stalks to edge-dependent encoding.

**Results.** Table 11 reports retention across all settings. Two patterns are consistent across datasets and perturbation types:

1. **SPD-Sheaf exhibits lower retention than Euclidean SheafNN under the matched architecture.** For example, on BACE under Drop with $p = 0.5$, SPD-Sheaf retains $69.5\%$ of its clean ROC-AUC while Euclidean SheafNN retains $83.0\%$; the gap of $\sim 13$ points is attributable solely to the SPD-versus-vector stalk choice.
2. **The gap widens with $p$.** As more edges are corrupted, SPD-Sheaf loses proportionally more performance, consistent with the hypothesis that it encodes richer information through edge-specific transformations.

This is the intended mechanistic signature: SPD-Sheaf's gains on clean graphs originate from *using edges more*, not from architectural factors orthogonal to the sheaf framework.

The value of this diagnostic lies in relative sensitivity across models under matched architecture, not in absolute retention numbers. Indiscriminate edge corruption is not a realistic deployment shift, and we do not claim SPD-Sheaf is more robust to adversarial or natural edge noise. Notably, BBBP under Rewire with $p = 0.1$ yields GCN retention exceeding $100\%$, indicating that GCN's predictions on BBBP depend only weakly on graph topology—which further supports the relative interpretation of this analysis.

## J. Computational Complexity

We analyze the time and space complexity of our SPD Sheaf Neural Network. Let $N$ denote the number of nodes, $E$ the number of edges, $d$ the dimension of SPD matrices (fixed at $d = 3$), $F$ the Euclidean feature dimension, and $L$ the number of layers.

**Equivariant Local Frames** We construct a per-node equivariant orthogonal frame $M_v \in \mathbb{R}^{3 \times 3}$ from coordinates via edge aggregation and cross products. This costs $\mathcal{O}(E + N)$ for $d = 3$.

**Per-Layer Time Complexity.** Each layer of our model consists of six main computational components:

1. **SPD Transform:** The transformation $\tilde{X}_v = Q_v X_v Q_v^\top$ involves: (i) learning orthogonal matrix $Q_v$: $\mathcal{O}(Nd^3)$. (ii) matrix multiplications on $N$ SPD matrices: $\mathcal{O}(Nd^3)$. Total: $\mathcal{O}(Nd^3)$.
2. **Sheaf Learner:** Learning orthogonal restriction maps from node features: (i) feature concatenation for each edge: $\mathcal{O}(EF)$; (ii) MLP forward pass producing skew-symmetric parameters: $\mathcal{O}(EF^2)$ (width scales with $F$); (iii) Cayley transform $(I - S/2)^{-1}(I + S/2)$ for $2E$ matrices: $\mathcal{O}(Ed^3)$. Total: $\mathcal{O}(EF^2 + Ed^3)$.
3. **Sheaf Laplacian Diffusion:** The sheaf-based message passing comprises: (i) congruence actions $M_{ve} X_v M_{ve}^\top$ for source and destination: $\mathcal{O}(Ed^3)$; (ii) logarithmic maps on edge SPD matrices: $\mathcal{O}(Ed^3)$; (iii) pullback transforms $M^\top(\cdot)M$: $\mathcal{O}(Ed^3)$; (iv) scatter-add aggregation: $\mathcal{O}(Ed^2)$; (v) eigenvalue normalization: $\mathcal{O}(Nd^3)$; (vi) exponential map: $\mathcal{O}(Nd^3)$. Total: $\mathcal{O}(Ed^3 + Nd^3)$.
4. **Euclidean Feature Propagation:** GraphSAGE with mean aggregation: $\mathcal{O}(EF + NF^2)$.

5. **Cross-Manifold Interaction:** The SPD-to-feature interaction: (i) log map to tangent space: $\mathcal{O}(Nd^3)$; (ii) linear projections and attention computation: $\mathcal{O}(Nd^2F + NF^2)$. Total: $\mathcal{O}(Nd^3 + Nd^2F + NF^2)$.
6. **SPD Nonlinearity:** Eigendecomposition and reconstruction: $\mathcal{O}(Nd^3)$.

**Overall Complexity.** Combining all six components, the per-layer complexity is:

$$\mathcal{O}\big(Nd^3 + Ed^3 + EF + EF^2 + NF^2 + Nd^2F\big).$$

Since the SPD dimension $d = 3$ is a fixed constant, this simplifies to:

$$\mathcal{O}(N + E + EF + EF^2 + NF^2 + NF) = \mathcal{O}(EF^2 + NF^2),$$

which is *linear* in both the number of nodes $N$ and edges $E$ for a fixed feature dimension $F$. The total complexity for $L$ layers is therefore $\mathcal{O}(LF^2(E + N))$.

**Graph-Level Pooling.** For graph classification:
- Power-Euclidean SPD pooling: $\mathcal{O}(Nd^3)$;
- Standard mean/sum pooling for Euclidean features: $\mathcal{O}(NF)$.

**Classification Head.** The factorized bilinear pooling (FBP) fusion requires $\mathcal{O}((d^2 + F)r)$ per sample, where $r$ is the factorization rank. The cross-attention variant has complexity $\mathcal{O}(F^2)$ per sample.

**Space Complexity.** The model stores the following intermediate activations per layer:
- Node SPD matrices: $\mathcal{O}(Nd^2)$
- Edge restriction maps: $\mathcal{O}(Ed^2)$
- Node features: $\mathcal{O}(NF)$

Total: $\mathcal{O}(Nd^2 + Ed^2 + NF)$. With $d = 3$ constant, this reduces to $\mathcal{O}(N + E + NF) = \mathcal{O}(NF + E)$.

The number of learnable parameters scales as $\mathcal{O}(LF^2)$.

**Comparison with Standard GNNs.** Compared to standard message-passing networks with $\mathcal{O}(EF + NF^2)$ complexity, our model has two additional sources of overhead:

1. **SPD manifold operations:** $\mathcal{O}((N + E)d^3)$ for matrix logarithms, exponentials, and congruence actions.
2. **Sheaf learner:** $\mathcal{O}(EF^2)$ for learning edge-specific restriction maps, compared to $\mathcal{O}(EF)$ for standard edge-wise operations.

The dominant terms are thus $\mathcal{O}(EF^2 + NF^2)$ versus $\mathcal{O}(EF + NF^2)$ for standard GNNs. In practice, for typical molecular graphs with $N \approx 20$–$50$ atoms and $F = 128$, the $NF^2$ term dominates both models, making the additional $EF^2$ overhead modest. The SPD operations with $d = 3$ contribute negligible cost relative to feature propagation.

**Numerical Precision.** Our implementation uses `float64` precision for SPD manifold operations (matrix logarithms, exponentials, and eigendecompositions). This choice is motivated by numerical stability: the composition of matrix exponentials and logarithms in sheaf convolution can produce gradient values that overflow or underflow in `float32`, leading to NaN during backpropagation.

# K. Future Works

The frame of this work mainly focuses on SPD manifold-based sheaf neural networks, and the future potential developments lie primarily in two areas. One relates to the relationship between restriction maps and the structure of the graph. The restriction map distinguishes and compares the initial information of different nodes, with its discrimination effectiveness being strongly influenced by the definition of the restriction map and the information transmission properties on the graph. One simple but meaningful question is whether the effectiveness of the restriction map can be controlled based on the given structure of the graph, thereby enhancing the overall efficiency of the sheaf neural network? Within the scope of this paper, we break it down into addressing the following works: (1) find the criterion for determining the "optimal", and (2) for a given graph and an appropriate restriction map defined on it, determine whether this definition of the restriction map is "optimal".

Another side is mostly focused on the mathematical framework itself. Our work primarily addresses the problem of inputting SPD-type data as features, while the underlying sheaf algebra structure still holds potential for further mathematical

refinement. People can further replacing sheaf types with other manifolds, or introducing more complex sheaf structures. Establishing connections between this more general sheaf theory and neural network performance would represent a significant advancement in developing sheaf neural network methodologies.

