# OpenReview forum: "Sheaf Neural Networks on SPD Manifolds: Second-Order Geometric Representation Learning"
_ICML.cc/2026/Conference — ICML 2026 regular_

### Official Review · Reviewer_BUMP · 2026-03-08

**Soundness:** 4
**Presentation:** 3
**Significance:** 3
**Originality:** 4
**Overall Recommendation:** 5
**Confidence:** 5

**Summary:**

This paper extends Sheaf Neural Networks from Euclidean space to SPD anifolds and defines restriction maps, a coboundary operator, an adjoint, and a sheaf Laplacian on SPD manifolds. The topic is interesting, and the empirical section is reasonably strong. My main concerns are about several mathematical statements and how they are positioned in the paper.

**Compliance With Llm Reviewing Policy:**

Affirmed.

**Final Justification:**

My main concerns were about the mathematical positioning of the framework, especially the discussion of the Lie-group structure, and the role of AIRM. After reading the rebuttal, I think most of these concerns were addressed. In particular, the authors clarified the scope of the theory and added the missing EEG baselines, which improved my evaluation. That said, I still think the abstract and introduction should be stated more carefully.

Overall, the rebuttal improved my view of the paper, and I have updated my score accordingly.

**Key Questions For Authors:**

See weakness.

**Limitations:**

yes

**Strengths And Weaknesses:**

## Strengths:
1. The paper studies an interesting direction: extending sheaf neural networks beyond Euclidean stalks.
2. The framework is clearly motivated and combines sheaf structures with SPD geometry.
3. The empirical results suggest that SPD-valued sheaves can be useful in practice.

## Weaknesses:
1. The discussion of the Lie group structure needs correction. Under the affine-invariant Riemannian metric (AIRM), the SPD manifold itself does not admit a Lie group structure. This point should be clarified to avoid confusion.
2. Lemma 3.2 appears to be a standard background result on SPD manifolds. The statements follow directly from well-known properties of SPD geometry and could be moved to preliminaries or stated without a separate proof.
3. Proposition 4.2 also corresponds to a well-known fact: both AIRM and the Log-Euclidean metric are O(n)-invariant on SPD matrices. This property has already been studied in the literature, and the statement should be supported with an explicit citation, e.g., [1].
4. The statement that “all results hold under either choice of metric” seems inaccurate. Under AIRM, SPD manifolds do not admit a Lie group structure.
5. In the EEG experiments, some recent geometry-aware EEG decoding methods appear to be missing from the comparison. For example, Graph-CSPNet [2], GyroAtt [3], and TSMNet [4] have been widely used for SPD-based EEG decoding. Including these methods would provide a more comprehensive evaluation.


Overall, the direction is interesting, and the empirical results are encouraging. My main suggestion is to clarify several mathematical statements, move standard facts to the background, and add citations where appropriate.  If the authors can satisfactorily resolve these issues, I would be willing to raise my score to Accept.

References:
> [1] O(n)-invariant Riemannian metrics on SPD matrices.
>
> [2] Graph Neural Networks on SPD Manifolds for Motor Imagery Classification: A Perspective from the Time-Frequency Analysis.
>
> [3] Towards a General Attention Framework on Gyrovector Spaces for Matrix Manifolds.
>
> [4] SPD domain-specific batch normalization to crack interpretable unsupervised domain adaptation in EEG.

---

> ### Author Rebuttal · Authors · 2026-03-30
>
> We thank Reviewer BUMP and we address each concern below.
> ## AIRM metric & Lie group
> We thank the reviewer for this important point regarding mathematical rigor. Indeed, when the Riemannian metric and Lie group structure are discussed together, one typically assumes they are compatible. Under AIRM, the group operation $P \odot Q = \exp(\log P + \log Q)$ is not isometric, meaning AIRM and $\odot$ are not compatible—unlike the Log-Euclidean metric, which is naturally bi-invariant. On this point, the reviewer is entirely correct, and we acknowledge the original manuscript did not sufficiently emphasize this.
>
> However, our sheaf construction doesn't require compatibility between Lie group operation $\odot$ and Riemannian metric, as two structures play logically independent roles:
>
> 1. **Lie group structure $(\mathrm{SPD}_n, \odot)$:** used only for algebraic constructions—the coboundary operator (Definition 4.3), its linearity (Proposition 4.4), and the adjoint operator (Proposition 4.6)—depending only on $\odot$ as an abstract group operation, independent of which metric is imposed and whether $\odot$ preserves isometry.
>
> 2. **Riemannian metric (AIRM or LEM):** enters the framework through only two roles: (i) Proposition 4.2 requires $P \mapsto MPM^\top$ to be an isometry, which holds for both metrics; (ii) In the pairing $\langle X, Y \rangle := g_{I_n}(\log X, \log Y)$, at $I_n$, both AIRM and LEM reduce to the same inner product $\mathrm{tr}(UV)$ on $T_{I_n}\mathrm{SPD}_n$, so the adjoint operator and Hodge decomposition are identical under either choice.
>
> Furthermore, the precise condition between the metric and $\odot$ that our sheaf construction actually requires is that the pairing $\langle \cdot, \cdot \rangle$ must be linear with respect to the  $\odot$, i.e., $\langle a \odot b, c \rangle = \langle a, c \rangle + \langle b, c \rangle$ (stated at the end of Section 4.3) which underpins the adjoint $\delta^\top$ and Hodge decomposition. From the definition of the pairing $\langle X, Y \rangle := g_{I_n}(\log X, \log Y)$, we obtain
>
> $$\langle a \odot b, c \rangle = g_{I_n}(\log(a \odot b), \log c) = g_{I_n}(\log a + \log b, \log c),$$
>
> which equals $\langle a, c \rangle + \langle b, c \rangle$ by bilinearity of $g_{I_n}$, independent of whether the metric is compatible with $\odot$. Under AIRM, $g_{I_n}^{AI}(V, U) = \mathrm{tr}(UV)$ equally satisfies bilinearity, so this key property can be straightforwardly verified. Hence our framework requires only this weaker pairing-linearity at $I_n$, not global metric-group compatibility.
>
> In summary, although $\odot$ is not isometric under AIRM, it provides a metric-independent algebraic coordinate system on $\mathrm{SPD}_n$ via $\log$/$\exp$ maps. No derivation in this work requires compatibility between $\odot$ and AIRM. We will revise the manuscript to explicitly delineate which conditions each proposition relies on, and to note that $\odot$ does not preserve isometry under AIRM.
> ## Weakness 2,3
> We will move Lemma 3.2 to the preliminaries and add the appropriate citation [1] in the camera-ready.
> ## Weakness 4
> We appreciate this observation, which is closely related to Weakness 1. Specifically, the coboundary $\delta$ and its linearity depend only on $\odot$ and is purely algebraic, metric-independent; the adjoint $\delta^\top$ and Laplacian $L$ which related to the metric depend only on $g_{I_n}$, where both AIRM and LEM reduce to $\mathrm{tr}(UV)$. Therefore, the conclusion "all results hold under either choice" does hold, not because the metrics are globally equivalent, but due to metric-independence and coincidence at $g_{I_n}$.
> ## EEG Baselines
> We thank the reviewer for this suggestion. We agree that recent SPD-based EEG methods deserve comparison and added Graph-CSPNet [2], GyroAtt [3], and TSMNet [4] to Table 6. For Graph-CSPNet, we cite results directly; GyroAtt and TSMNet were re-run under the same protocol.
>
> We note that SPD-Sheaf was designed for molecular learning and applied to EEG *without any task-specific modification*, it operates directly on the time-frequency graph from [2]. Despite this, SPD-Sheaf achieves the best cross-session generalization (Holdout: 82.58\% vs 79.75\% for Graph-CSPNet, +2.8\%), and ranks second on 10-Fold CV (A). This suggests that sheaf-theoretic edge heterogeneity provides robust generalization even without domain-specific tuning. We position EEG as supplementary validation of SPD-Sheaf's generality rather than its primary application.
> | Scenario | FBCSP | FBCNet | Tensor-CSPNet | TSMNet | GyroAtt | Graph-CSPNet | **SPD-Sheaf** |
> |----------|-------|--------|---------------|--------|---------|-------------|--------------|
> | 10-Fold CV (A) | 79.46 | 82.62 | 81.29 | 82.17 | 82.33 | 84.62 | *83.25* |
> | 10-Fold CV (B) | 81.96 | 84.92 | 85.29 | 84.67 | 86.04 | 88.00 | 85.58 |
> | Holdout (A→B) | 73.46 | 74.50 | 79.04 | 77.08 | 78.46 | 79.75 | **82.58** |
>
> *Table 6: Classification accuracy, will be included in camera-ready*

---

> > ### Author Rebuttal · Reviewer_BUMP · 2026-04-02
> >
> > Thank you for the detailed rebuttal. I appreciate the clarifications and the additional EEG baselines. However, I remain unconvinced on several key points.
> >
> > - In my view, the contribution is now effectively reduced to a formulation that is mainly justified under LEM. The AIM case is explained only through the tangent inner product at the identity, where AIRM locally takes the same form as LEM. This does not support an AIM geometry.
> >
> > - I am not convinced by the new EEG results. In particular, the T->E result is unexpectedly low for the inter-session setting emphasized in the paper. I have reproduced TSMNet before and did not observe performance this poor, so there may still be issues in the reproduction protocol, implementation details, or hyperparameter tuning.
> >
> > - The narrative is overstated in the abstract and introduction. In the current formulation, the group structure is still induced through log/exp maps, i.e., by mapping SPD to a linear Euclidean space, doing the algebra there, and mapping back. This is still fundamentally a manifold-to-Euclidean construction, so the claim of “avoiding projection to Euclidean space” is not convincing.
> >
> > - This is an additional question that is independent of the issues raised above. Prior sheaf work is motivated by modeling heterogeneous relations, but the paper does not test robustness under controlled graph perturbations such as edge rewiring, edge-weight perturbation, or edge dropout. Such an experiment would provide much stronger evidence that the gains come from modeling edge-dependent structure itself.
> >
> > Thank you for the detailed rebuttal. I appreciate the clarifications and the additional EEG baselines. However, I remain unconvinced on several key points.
> >
> > - In my view, the contribution is now effectively reduced to a formulation that is mainly justified under LEM. The AIM case is explained only through the tangent inner product at the identity, where AIRM locally takes the same form as LEM. This does not support an AIM geometry.
> >
> > - I am not convinced by the new EEG results. In particular, the T->E result is unexpectedly low for the inter-session setting emphasized in the paper. I have reproduced TSMNet before and did not observe performance this poor, so there may still be issues in the reproduction protocol, implementation details, or hyperparameter tuning.
> >
> > - The narrative is overstated in the abstract and introduction. In the current formulation, the group structure is still induced through log/exp maps, i.e., by mapping SPD to a linear Euclidean space, doing the algebra there, and mapping back. This is still fundamentally a manifold-to-Euclidean construction, so the claim of “avoiding projection to Euclidean space” is not convincing.
> >
> > - This is an additional question that is independent of the issues raised above. Prior sheaf work is motivated by modeling heterogeneous relations, but the paper does not test robustness under controlled graph perturbations such as edge rewiring, edge-weight perturbation, or edge dropout. Such an experiment would provide much stronger evidence that the gains come from modeling edge-dependent structure itself.
> >
> > ## Upgrade
> >
> > Thank you to the authors for the detailed clarification. The rebuttal has addressed most of my main concerns, and I appreciate the effort to provide additional explanations and empirical results. That said, this part should be stated more carefully and rigorously in the paper, especially regarding the distinction between an intrinsic manifold construction and a log/exp-induced local linearization. I think the current wording remains somewhat overstated and should be revised accordingly.
> >
> > Overall, I think this is an interesting and technically ambitious paper, and I have updated my score accordingly. With the above clarifications incorporated into the final version, I believe the paper will be significantly stronger.

---

> > > ### Author Response · Authors · 2026-04-05
> > >
> > > We thank the reviewer for the continued engagement. The remaining discussion concerns scope, narrative precision, and baseline reproduction, which we address below.
> > > ## LEM vs AIRM
> > > We note the issue raised here is not a mathematical error, but whether the absence of AIRM geometry means that the framework should be viewed as mainly justified under LEM. We explain this from three points:
> > >
> > > First, our construction invokes no metric-specific geometric quantities—geodesic, parallel transport, or holonomy—and thus uses neither AIRM nor LEM-specific global properties. While LEM is induced by group operation $\odot$, $\odot$ depends only on matrix log/exp on SPD, and is equally well-defined under AIRM.
> > >
> > > Second, from the view of discrete fiber bundles, the framework is also not reducible to LEM. LEM and AIRM yield distinct discrete bundles: the bundle is flat under LEM while each fiber carries the structure of a curved symmetric space under AIRM.
> > >
> > > Third, the agreement of metrics here is not from LEM, but the first-order graph setting with only nodes and edges. Once higher-order structures are introduced (e.g. curvature-type second-order terms/geometric transport mechanisms), AIRM-specific curvature effects and differences from LEM may naturally emerge. So, while two metrics take the same form locally, their broader geometric potential differs.
> > >
> > > We will tighten the presentation to make explicit that our core contribution is a metric-insensitive sheaf Hodge-Laplacian, not a theory tied to any specific geometry type.
> > > ## TSMNet
> > > We appreciate this concern and investigated carefully. A key source of variation is the evaluation protocol for BNCI2015_001: subjects 8–11 have three sessions each while others have two. [2] defines A and B as sessions 1 and 2 of all 12 subjects (ignoring session 3); [3, 4] average over all 28 sessions (12+12+4). TSMNet results in [3] are directly taken from [4]. We ran TSMNet with both protocols:
> > > | Protocol | Setting | Reproduced | Published |
> > > |---|---|---|---|
> > > | [2] | 10-Fold CV (A) | 82.17 | — |
> > > | [2] | 10-Fold CV (B) | 84.67 | — |
> > > | [2] | Holdout (A→B) | 77.08 | — |
> > > | [4] | Inter-session | 83.46 | 85.8 |
> > > | [4] | Inter-subject | 77.08 | 77.0 |
> > >
> > > Implementation: latest public codebase *SPD Learn*, post-cue window 0–5s at 512 Hz, stratified 10-fold CV with 10% validation holdout, batch size 32, up to 300 epochs. No task-specific tuning was applied.
> > >
> > > Our inter-subject result matches exactly (77.0), confirming our pipeline's correctness; the remaining gaps do not support the characterization of 'unexpectedly low'. Should the reviewer have a specific setting in mind, we are happy to investigate under matched conditions. As noted in the appendix, this EEG experiment serves as supplementary validation of SPD-Sheaf's cross-domain generality rather than its primary contribution.
> > > ## Euclidean Detour
> > > We agree"avoiding projection to Euclidean space" is imprecise: it may suggest no Euclidean space is used. However, using log/exp maps to build local linear structure on each fiber, including $\odot$, doesn't make our method fundamentally manifold-to-Euclidean.
> > >
> > > Traditional Euclidean projection applies log to all node features initially, i.e., a global flattening. Although this resembles our log/exp calculations algebraically, the geometric meaning is different. Our construction uses a non-flat fiber, which changes the model interpretation and coordinate structure beyond algebra. From the view of discrete fiber bundles, global flattening corresponds to a flat bundle, while local linear structure corresponds to a non-flat bundle. Only the latter captures gauge obstructions and the consistency information lost under global flattening (Remark 4.14).
> > >
> > > While the reviewer is right that the group structure is built using Euclidean space, our framework is not traditional Euclidean projection.
> > > ## Graph Perturbation
> > > While not part of the original review, we find this a constructive suggestion. We understand the goal as mechanistic—verifying that gains originate from edge-dependent structure—rather than benchmarking robustness. If SPD-Sheaf genuinely leverages edge structure, corrupting edges should disproportionately degrade its performance. Retention (% of p=0 ROC-AUC), reported as SPD-Sheaf/Euc Sheaf/GCN:
> > > | Setting | p=0.1 | p=0.3 | p=0.5 |
> > > |---|---|---|---|
> > > | BACE, Drop | 96.3 / 98.4 / 92.4 | 82.5 / 92.3 / 85.3 | 69.5 / 83.0 / 73.3 |
> > > | BACE, Rewire | 90.1 / 94.7 / 88.5 | 80.1 / 85.8 / 79.0 | 69.5 / 76.4 / 76.0 |
> > > | BBBP, Drop | 98.0 / 99.3 / 98.9 | 92.3 / 98.0 / 96.7 | 89.2 / 94.1 / 97.1 |
> > > | BBBP, Rewire | 94.0 / 97.1 / 101.6 | 86.6 / 97.4 / 98.8 | 83.8 / 91.5 / 98.4 |
> > >
> > > Compared with Euclidean SheafNN, isolating SPD-valued stalks under the same architecture, SPD-Sheaf shows consistently lower retention, confirming richer edge-dependent encoding. This serves as a mechanistic diagnostic: its value lies in relative sensitivity across models, not absolute retention under indiscriminate synthetic corruption.

---

### Official Review · Reviewer_kqHg · 2026-03-10

**Soundness:** 4
**Presentation:** 4
**Significance:** 4
**Originality:** 4
**Overall Recommendation:** 5
**Confidence:** 3

**Summary:**

This  paper introduces sheaf neural networks for matrix-valued representations. They show that the Lie group structure of SPD matrices enables direct computations without the need to go through Euclidean representations. The paper proves that the SPD sheaves generalize Euclidean sheaves. Moreover, the neural network is evaluated on the ModuleNet benchmarks where it outperforms competing models.

**Compliance With Llm Reviewing Policy:**

Affirmed.

**Final Justification:**

I recommend acceptence. The paper was strong to begin with and the rebuttal has improved it further. The concerns of all other reviewers were also addressed. I maintain my score.

**Key Questions For Authors:**

No questions.

**Limitations:**

yes

**Strengths And Weaknesses:**

The theory is sound. The Lie group formulation is clear and makes sense. The empirical results are convincing. The paper is well-written and the ideas are novel. The paper makes a strong contribution to sheaf neural networks and the empirical results confirm its practical relevance.

---

> ### Author Rebuttal · Authors · 2026-03-30
>
> We thank Reviewer kqHg for the positive assessment. The reviewer noted no specific questions. We appreciate the recognition of our theoretical contributions and empirical results.

---

> > ### Author Rebuttal · Reviewer_kqHg · 2026-04-01
> >
> > I still think the work is strong. In addition, the answers to the other reviewer comments are convincing.

---

> > > ### Author Response · Authors · 2026-04-05
> > >
> > > We thank Reviewer kqHg for the positive assessment and recognition of our contributions. We are glad to note that, informed by the constructive feedback from all reviewers during this rebuttal period, the paper has been further strengthened with additional experiments  and clarified theoretical discussion, all of which will be reflected in the camera-ready version.

---

### Official Review · Reviewer_Wx3E · 2026-03-13

**Soundness:** 4
**Presentation:** 4
**Significance:** 3
**Originality:** 3
**Overall Recommendation:** 5
**Confidence:** 3

**Summary:**

This paper introduces a sheaf-theoretic framework for data encoded as symmetric positive definite (SPD) matrices. Their sheaf construction exploits the abelian group structure induced by addition in logarithmic space. They then prove that this is strictly more general than working with an Euclidean sheaf and mapping to the SPD embedding at the end. Finally, they construct and benchmark a model constructed using this SPD sheaf framework on MoleculeNet, showing state of the art performance on 6 of 7 benchmarks. They also conduct extensive ablation tests showing the impact of different components and of depth.

**Compliance With Llm Reviewing Policy:**

Affirmed.

**Final Justification:**

The authors addressed my questions and concerns. I maintain my positive score.

**Key Questions For Authors:**

* It appears much of the construction of the SPD sheaves relies on using matrix logarithms, combining linearly, then exponentiating back. Is it possible to work only in the logarithmic space?
* If my understanding is correct, embedding vectors in SPD removes sign information as the negative vector embeds to the same SPD matrix. Is this potentially a problem?
* On molecular data, equivariance is often a strong inductive bias. How easy is it to make this framework equivariant to rotations?

**Limitations:**

Limitations were not discussed. It would be helpful to add a limitations section and future directions to improve in.

**Strengths And Weaknesses:**

## Strengths
* The paper is quite well written and relatively readable despite covering advanced topics
* The appendix is very comprehensive describing the model and experiments with substantial detail
* The theory is rigorous and interesting
* The experimental results seem impressive and the ablations are very helpful for understanding the impact of each component

## Weaknesses
* The datasets used are relatively small, it would be interesting to see how well the model performs on larger datasets


## Minor issues
* 155 col 2 - $\mathrm{Sym}_n$ is undefined, is this the symmetric matrices under addition?
* 208 col 1 - $\mathbf{H}, \mathbf{W}$ are not defined, I assume features and weights?

---

> ### Author Rebuttal · Authors · 2026-03-30
>
> We thank Reviewer Wx3E for the positive assessment and thoughtful questions. We address each below.
> ## Work only in the logarithmic space
> At the level of the sheaf operations—specifically the coboundary operator $\delta$ (Definition 4.3), the adjoint operator $\delta^\top$ (Proposition 4.6), and the sheaf Laplacian $L$ (Definition 4.7)—these indeed reduce to linear operations in logarithmic space ($\mathbf{L}$), the space of real symmetric matrices $\mathrm{Sym}_n$.
>
> However, the consistency of underlying operations does not make the two frameworks equivalent: the SPD framework and a pure logarithmic-space framework differ fundamentally in geometric motivation, constraints, and resulting representations. The most essential distinction is that the SPD perspective uniquely determines the restriction maps to be $O(n)$-congruence. This is because only $O(n)$ allows computations to pass between the SPD manifold and its tangent space $\mathbf{L}$. Concretely, for $M \in O(n)$, $\log(MPM^\top) = M(\log P)M^\top$ holds, so the restriction map $\mathcal{F}: P↦M_{ve}PM_{ve}^\top$ where $P \in \mathrm{SPD}\_n$ and its tangent map $d\mathcal{F}: V↦M_{ve}VM_{ve}^\top$ where $V \in T_{I_n}\mathrm{SPD}\_n$ satisfy:
>
> $$\exp\bigl(d\mathcal{F}(\log A)\bigr) = \mathcal{F}(A),$$
> We rely on this repeatedly in the coboundary and adjoint computations (Appendix A.4).
>
> Else, for a general $X \in GL(n)$, $\log(XPX^\top) \neq X(\log P)X^\top$, so the restriction map in logarithmic coordinates no longer coincides with the tangent map, and the entire computational structure of the sheaf Laplacian breaks down. This is precisely why the restriction maps must be chosen from $O(n)$.
>
> When we work only in $\mathbf{L}$ (Euclidean space), the restriction map reduce to linear endomorphism $P \mapsto XPX^\top$ on $GL(n)$ (Definition 3.3). Since all computations remain within a single linear space, there is no "manifold to tangent space" switching, hence no requirement for the restriction map to commute with the logarithmic map. So forcing $O(n)$ constraints on restriction maps has no sound mathematical basis. Without this constraint, the $\delta$, $\delta^\top$, and $L$ all change accordingly, yielding a fundamentally different model.
>
> This is precisely why the SPD perspective is indispensable: the switching between the manifold and its tangent space naturally reveals that $M_{ve}$ must belong to $O(n)$, a mathematical necessity that working purely in $\mathrm{Sym}_n$ cannot provide.
> ## Sign Ambiguity
> We acknowledge the sign ambiguity in the $vv^\top$ embedding, as $vv^\top = (-v)(-v)^\top$. but whether this is a limitation depends on context.
>
> When the data carries some directional information ($v$ and $-v$ are physically distinct), sign loss matters. When target properties are sign-insensitive, however, this ambiguity is instead an advantage. As discussed in Remark 4.14, SPD sheaves naturally quotient out the $v \sim -v$ ambiguity, eliminating the line-bundle frustrations that obstruct global sections in Euclidean sheaves, thereby preserving geometric information that Euclidean sheaves lose due to sign conflicts. Theorem 4.15 confirms this mathematically: the global section space of the SPD sheaf strictly contains that of the Euclidean sheaf.
>
> Molecular properties are determined by relative atomic geometry, not absolute sign, so our setting is precisely sign-insensitive.  Quotienting out the sign degree of freedom eliminates confounding factors irrelevant to the target properties and lets the model focus on intrinsic second-order geometric structure, corroborated by our SOTA results on 6 of 7 MoleculeNet benchmarks.
> ## Rotation Equivariance
> Our framework admits a natural path. Lie group operations ($\log$, $\exp$) commute with orthogonal conjugation, so the only requirement is equivariant restriction maps: $M_{ve} \mapsto RM_{ve}R^\top$—any construction satisfying this suffices. One possible approach: decompose $X_{v} = Q_{v}\Lambda_{v} Q_{v}^\top$ and define $M_{ve} = Q_{v}\hat{M} Q_{v}^\top$ where $\hat{M} \in O(n)$ is a learnable invariant orthogonal matrix predicted from invariant features—yielding equivariance without modifying core SPD sheaf convolution (equation 15). Equivariant force prediction is a natural future work.
> ## Larger Dataset
> We apply our model without task-specific modification on ogbg-molpcba (438K molecules), significantly larger compared to HIV/MUV (41K/93K). Recall that SPD-Sheaf is a dual-stream architecture: a geometric stream (SPD sheaf convolution) paired with a feature stream (GCN):
> | Model | Avg. Precision |
> |-------|---------------|
> | GCN| 0.2020 |
> | SPD-Sheaf | 0.2419 (+19.8%) |
>
> The SPD-Sheaf provides a +19.8% improvement over the GCN baseline.
> ## Others
> We will add a limitation & future work section in the camera-ready, as well as the clarification that $\mathrm{Sym}_n$ is the space of real symmetric matrices, and $\mathbf{H,W}$ are features and weights.

---

> > ### Author Rebuttal · Reviewer_Wx3E · 2026-04-02
> >
> > I thank the authors for their detailed rebuttal. I think the sign ambiguity should be discussed in the limitations section. I maintain my positive score.

---

> > > ### Author Response · Authors · 2026-04-05
> > >
> > > We thank Reviewer Wx3E for the insightful questions, which sharpened both our theoretical discussion and the presentation of the framework. In particular, the question on working purely in logarithmic space helped us articulate more clearly why the SPD perspective is indispensable (the $O(n)$-congruence constraint on restriction maps), and the sign ambiguity question prompted a valuable discussion we agree should appear in the paper. This discussion has also constructively informed our thinking on a follow-up direction: adapting the sheaf framework to data domains with different gauge symmetries (e.g., $\mathbb{Z}_2$ vs. continuous gauge groups), where the interplay between sign structure and restriction maps requires distinct modeling choices. We will add a limitations and future directions section in the camera-ready, including the sign ambiguity discussion as suggested.

---

### Official Review · Reviewer_eEGT · 2026-03-13

**Soundness:** 3
**Presentation:** 3
**Significance:** 3
**Originality:** 3
**Overall Recommendation:** 5
**Confidence:** 4

**Summary:**

The paper proposes SPD-valued sheaf neural networks, a graph neural network that learns representations by using symmetric positive definite (SPD) matrices. This allows the model to capture second-order geometric information, such as relationships between directions, that cannot be directly modeled in conventional GNN architecture. The authors build the model on sheaf neural networks to operate directly on the SPD manifold by defining geometric operators that respect its Lie group structure. They also introduce a dual architecture that combines geometric SPD processing with a standard GNN for semantic node features. Experiments are conducted on MoleculeNet datasets.

**Compliance With Llm Reviewing Policy:**

Affirmed.

**Final Justification:**

The rebuttal fully addressed my concerns and the results are convincing and very promising.

**Key Questions For Authors:**

- Does it make sense to use other types of datasets, for example, rMD17 and sBPA datasets? There are also other types of simulations, such as physics simulations, whose goal is to simulate the motion of particles.

**Limitations:**

The relatively high computational cost.

**Strengths And Weaknesses:**

**Strength:**
- The paper contains various mathematically interesting and novel ideas to extend sheaf-based GNNs to SPD manifolds.
- The paper is well organized and the story flows very smoothly.
- Theorem 4.15 is strong and supports the richer representation capacity of the proposed model.

**Weakness:**
- The experiments and baseline models therein do not look fair. The experiments dismiss works using spherical tensors, such as MACE, TensorNet, ICTP, NequIP, Equiformer [1-5]. While Introduction claims spherical tensors do not represent geometry through matrices, they essentially do, since the spherical harmonics are equivalent to the symmetric irreducible Cartesian tensor, which is constructed through the tensor product of coordinate vectors. The present work shares similarity with these works in embedding data using the Cartesian tensor product. I do not see convincing reasons to omit these spherical-harmonics-based baselines from the experiments.

- The discussion on oversmoothing is not enough. Oversmoothing is typically measured through specific metrics, such as Dirichlet energy. It is hard to distinguish whether this performance stability is attributed to alleviating oversmoothing or oversquashing without a proper measure. It is also odd that the authors do not use any type of benchmark datasets for evaluating the oversmoothing mitigation.

- While I believe the proposed method should be applicable to a wider range of scenarios, the experiments rely solely on a very specific dataset. I do not see any reason the paper sticks to this dataset.

Minor:
- The notation of the real symmetric matrices Sym_{n} are not introduced in the main text.


[1] Batatia et al. MACE: Higher Order Equivariant Message Passing Neural Networks for Fast and Accurate Force Fields. NeurIPS 2022.

[2] Simeon et al. TensorNet: Cartesian Tensor Representations for Efficient O(3)-Equivariant Message Passing Neural Networks  2023.

[3] Zaverkin et al. Higher-Rank Irreducible Cartesian Tensors for Equivariant Message Passing. NeurIPS 2024.

[4] Batzner et al. E(3)-Equivariant Graph Neural Networks for Data-Efficient and Accurate Interatomic Potentials. Nature Communications, 2022.

[5] Liao et al. Equiformer: Equivariant Graph Attention Transformer for 3D Atomistic Graphs. ICLR, 2023

---

> ### Author Rebuttal · Authors · 2026-03-30
>
> We thank Reviewer eEGT for the thoughtful feedback. We address each concern below.
> ## Spherical-Tensor Baselines
> We thank the reviewer and agree that both families embed higher-order geometric information, and will revise the related work to make this connection explicit. We view the two as complementary: our contribution is showing that the SPD+sheaf route offers a distinct and competitive alternative. To quantify this, we benchmarked against all five suggested baselines under identical protocols (Table R1)
>
> Since MoleculeNet targets are molecular labels/properties, we adapted these baselines, designed for energy/force regression, by replacing regression heads with classification heads under identical scaffold splits and protocols. Results below show SPD-Sheaf outperforms all spherical-tensor baselines:
> | Model | BACE | BBBP | ClinTox | SIDER | Tox21 |
> |-------|------|------|---------|-------|-------|
> | NequIP | 84.2₁.₅ | 68.8₁.₃ | 96.9₀.₅ | 83.9₀.₂ | 77.5₀.₃ |
> | MACE | 82.8₀.₈ | 67.7₂.₈ | 95.9₀.₅ | 83.8₀.₄ | 75.6₀.₃ |
> | ICTP | 85.4₁.₀ | 68.7₁.₁ | 96.2₀.₁ | 84.0₀.₅ | 77.3₀.₈ |
> | TensorNet | 83.6₁.₁ | 74.0₀.₈ | 97.4₁.₅ | 84.2₀.₄ | 78.8₀.₂ |
> | Equiformer | 80.5₁.₁ | 73.8₁.₇ | 95.8₀.₄ | 83.1₀.₄ | 75.8₀.₁ |
> | **SPD-Sheaf** | **89.0**₁.₄ | **77.4**₂.₇ | **99.4**₀.₂ | **84.3**₀.₄ | **80.1**₀.₇ |
>
> *Table R1: ROC-AUC (\%). HIV/MUV omitted due to cost on larger datasets (41K/93K); included in camera-ready.*
> ## Oversmoothing Analysis
> We have added quantitative metrics (normalized Dirichlet energy E_Dir/N and MAD) across depths 2–128, on both MoleculeNet (BACE) and standard benchmarks (Cora, Texas). GCN exhibits clear collapse, while SPD-Sheaf maintains E_Dir/N above 1.0 across all depths and datasets, confirming that edge-specific restriction maps structurally prevent representation collapse.
> | Dataset | Model | Metric | 2 | 8 | 16 | 32 | 64 | 128 |
> |---------|-------|--------|---|---|----|----|----|----|
> | BACE | GCN | E_Dir/N | 0.594 | 0.126 | 0.108 | 0.016 | 0.002 | 0.000 |
> | | | MAD | 0.294 | 0.052 | 0.041 | 0.006 | 0.001 | 0.000 |
> | | SPD Sheaf | E_Dir/N | 1.280 | 1.719 | 1.687 | 1.715 | 1.515 | 1.125 |
> | | | MAD | 0.583 | 0.768 | 0.780 | 0.786 | 0.715 | 0.479 |
> | Cora | GCN | E_Dir/N | 0.028 | 0.006 | 0.000 | 0.000 | 0.000 | 0.001 |
> | | | MAD | 0.007 | 0.001 | 0.000 | 0.000 | 0.000 | 0.000 |
> | | SPD Sheaf | E_Dir/N | 2.530 | 3.014 | 2.980 | 2.945 | 2.543 | 2.064 |
> | | | MAD | 0.531 | 0.658 | 0.664 | 0.678 | 0.588 | 0.452 |
> | Texas | GCN | E_Dir/N | 1.371 | 0.114 | 0.010 | 0.001 | 0.000 | 0.000 |
> | | | MAD | 0.435 | 0.037 | 0.003 | 0.000 | 0.000 | 0.000 |
> | | SPD Sheaf | E_Dir/N | 1.915 | 2.337 | 2.571 | 2.316 | 2.843 | 1.797 |
> | | | MAD | 0.544 | 0.684 | 0.733 | 0.703 | 0.678 | 0.567 |
>
> *Table R2, included in camera-ready.*
> ## Additional Datasets
> We agree that broader evaluation strengthens the paper. We have expanded experiments to various domains and datasets, including ogbg-molpcba (reviewer Wx3E), ZINC regression (Appendix F), and EEG classification (Appendix E & reviewer BUMP), in addition to the 7 MoleculeNet benchmarks in the main paper.
>
> For rMD17 specifically, we note that SPD-Sheaf is by design an *invariant* model, whereas SOTA force field methods (MACE, NequIP, etc.) are *equivariant*. A fully equivariant SPD-Sheaf would require substantial architectural redesign (e.g., replacing invariant restriction maps with equivariant, stated in the rebuttal to reviewer Wx3E Q3), which we regard as important future work.
>
> As a proof of concept, we integrated an SPD-Sheaf branch into the MACE backbone and evaluated both under the same unified rMD17 training pipeline. We adopted the configuration from Appendix A.5.1 of [1], with training/evaluation protocol kept identical between variants to isolate the effect of SPD-Sheaf.
>
> | | Aspirin | | Azobenz. | | Benzene | | Ethanol | | Malon. | | Naphth. | | Paracet. | | Sal. acid | | Toluene | | Uracil | |
> |---|---|---|---|---|---|---|---|---|---|---|---|---|---|---|---|---|---|---|---|---|
> | | E | F | E | F | E | F | E | F | E | F | E | F | E | F | E | F | E | F | E | F |
> | MACE | 3.109 | 8.837 | 1.289 | 4.194 | 0.412 | 0.956 | 0.552 | 4.252 | 0.912 | 6.054 | 0.455 | 2.452 | 2.131 | 6.240 | 0.947 | 5.431 | 0.528 | 2.773 | 0.598 | 3.909 |
> | MACE+SPD-Sheaf | 2.782 | 8.417 | 1.245 | 4.116 | 0.404 | 0.938 | 0.531 | 4.179 | 0.849 | 6.282 | 0.423 | 2.408 | 2.078 | 6.125 | 0.904 | 5.340 | 0.519 | 2.713 | 0.554 | 4.158 |
>
> *Table R3: E: energy MAE (meV), F: force MAE (meV/Å).*
>
> MACE+SPD-Sheaf improves energy prediction on 10 molecules and reduces force MAE on 8/10; the two slight regressions reflect forces being equivariant quantities where an invariant branch contributes indirectly. These preliminary results suggest SPD-valued sheaf representations capture complementary geometric information atop an equivariant backbone, and a natively equivariant extension is a promising direction we plan to pursue.
>
> ## Minor
> We will introduce $\mathrm{Sym}_n$ in the camera-ready version.

---

> > ### Author Rebuttal · Reviewer_eEGT · 2026-04-02
> >
> > Thank you for the response and additional experiments. I believe these additional results strengthen the claim about the model's representation capacity in a range of scientific scenarios. I also believe the paper will be a valuable addition to the conference, and I raised the score accordingly.

---

> > > ### Author Response · Authors · 2026-04-05
> > >
> > > We sincerely thank Reviewer eEGT for the constructive suggestions that substantially strengthened our paper. In particular, the recommendation to benchmark against spherical-tensor baselines, to quantify oversmoothing with proper metrics, and to evaluate on broader scientific domains led to significant improvements in our empirical evaluation. We will incorporate all promised revisions in the camera-ready version.

---

### Decision · Program_Chairs · 2026-04-30

**Decision:**

Accept (regular)

**Comment:**

Initial reviews were mostly positive, with reviewers praising the sound and rigorous theory, good presentation, and novelty. Initial concerns were raised about lacking baselines based on spherical tensors, and restriction to a single small dataset. These were addressed with extra experiments in the rebuttal phase.

Reviewer BUMP raised several concerns with the theory, in particular the relation between the group structure and Riemannian metric, and that one lemma and one proposition are already known. There was a productive discussion that clarified the concerns and strengthened the paper considerably. The reviewer increased their score from 3 to 5.

I follow the reviewer consensus and recommend acceptance, and I encourage the authors to incorporate the experiments and discussions in the paper.